# A CAF40-binding motif facilitates recruitment of the CCR4-NOT complex to mRNAs targeted by *Drosophila* Roquin

Annamaria Sgromo[1], Tobias Raisch[1], Praveen Bawankar[1], Dipankar Bhandari[1], Ying Chen[1], Duygu Kuzuoğlu-Öztürk[1], Oliver Weichenrieder[1] & Elisa Izaurralde[1]

Human (*Hs*) Roquin1 and Roquin2 are RNA-binding proteins that promote mRNA target degradation through the recruitment of the CCR4-NOT deadenylase complex and are implicated in the prevention of autoimmunity. Roquin1 recruits CCR4-NOT via a C-terminal region that is not conserved in Roquin2 or in invertebrate Roquin. Here we show that Roquin2 and *Drosophila melanogaster* (*Dm*) Roquin also interact with the CCR4-NOT complex through their C-terminal regions. The C-terminal region of *Dm* Roquin contains multiple motifs that mediate CCR4-NOT binding. One motif binds to the CAF40 subunit of the CCR4-NOT complex. The crystal structure of the *Dm* Roquin CAF40-binding motif (CBM) bound to CAF40 reveals that the CBM adopts an α-helical conformation upon binding to a conserved surface of CAF40. Thus, despite the lack of sequence conservation, the C-terminal regions of Roquin proteins act as an effector domain that represses the expression of mRNA targets via recruitment of the CCR4-NOT complex.

---

[1] Department of Biochemistry, Max Planck Institute for Developmental Biology, Spemannstrasse 35, Tübingen 72076, Germany. Correspondence and requests for materials should be addressed to E.I. (email: elisa.izaurralde@tuebingen.mpg.de).

The CCR4-NOT deadenylase complex plays a central role in bulk mRNA degradation by catalysing the removal of mRNA poly(A) tails, which is the first step in general mRNA decay[1]. In addition to its role in global mRNA degradation, the CCR4-NOT complex regulates the expression of a large number of specific mRNAs, to which it is recruited via interactions with RNA-associated proteins. Consequently, CCR4-NOT functions as a major downstream effector complex in posttranscriptional mRNA regulation in eukaryotes.

The CCR4-NOT complex consists of several structurally and functionally distinct modules, which assemble around the NOT1 scaffold subunit[1]. NOT1 contains several α-helical domains that provide binding surfaces for the individual modules. A central domain of NOT1 that is structurally related to the middle portion of eIF4G (termed the NOT1 MIF4G domain) provides a binding site for the catalytic module, which comprises two deadenylases, namely CAF1 or its paralogue POP2 (also known as CNOT7 and CNOT8, respectively, in humans), and CCR4a or its paralogue CCR4b (also known as CNOT6 and CNOT6L, respectively, in humans). The NOT1 MIF4G domain also serves as a binding platform for the DEAD-box protein DDX6 (also known as RCK), which functions as a translational repressor and decapping activator[2,3]. C-terminal to the MIF4G domain, NOT1, contains a CAF40/NOT9-binding domain, CN9BD, that binds to the highly conserved CAF40 subunit, which is also known as CNOT9 (refs 2,3), followed by a NOT1 superfamily homology domain SHD, which interacts with NOT2-NOT3 heterodimers to form the NOT module[4,5].

The CAF40 and NOT modules have no catalytic activity and have been implicated in mediating interactions with RNA-associated proteins that recruit the CCR4-NOT complex to their targets[2–10]. These proteins include the GW182 family, which is involved in miRNA-mediated gene silencing in animals[2,3], tristetraprolin (TTP), a protein required for the degradation of mRNAs containing AU-rich elements[6], the germline determinant Nanos[7–9] and the human Roquin1 and Roquin2 proteins[10].

The vertebrate Roquin proteins are negative regulators of T follicular helper cell differentiation and autoimmunity in vertebrates[11,12]. There are two partially redundant paralogues, Roquin1 and Roquin2 (initially named membrane-associated nucleic acid-binding protein), in vertebrates and only one family member in invertebrate species[11,12]. The proteins feature an N-terminal folded region followed by a C-terminal extension of variable length and low sequence complexity that is predicted to be predominantly unstructured[13,14] (Fig. 1a). The N-terminal region contains a RING finger E3 ubiquitin ligase domain, a ROQ RNA-binding domain flanked by a bilobed HEPN domain and a CCCH-type zinc finger domain, all of which are highly conserved in metazoans and define the protein family[14–20]. The RING domains of Hs Roquin2 and the Caenorhabditis elegans homologue of Roquin1, RLE-1, (regulation of longevity by E3) exhibit E3 ubiquitin ligase activity[21,22].

The ROQ domain of Hs Roquin1 and Roquin2 recognizes specific stem-loop structures in the 3′-untranslated region (UTR) of target mRNAs. These targets include mRNAs encoding regulators of inflammation such as the inducible T-cell costimulator, the costimulatory receptor Ox40, neuropilin-1, interleukin-6, interferon γ (IFN-γ) and the tumor necrosis factor-α (TNF-α)[10,14,16,23–31]. Hs Roquin1 and Roquin2 downregulate these mRNA targets through interactions with the CCR4-NOT deadenylase complex and decapping factors[10,32].

For Hs Roquin1, it has been shown that the interaction with the CCR4-NOT complex is mediated by the C-terminal region of the protein that is conserved only among vertebrate Roquin1 orthologues[10]. However, it is not known how Roquin2 recruits the CCR4-NOT complex, because its C-terminal region shows no similarity with that of Roquin1. In addition, the C-terminal regions of the invertebrate Roquin proteins are highly divergent[18,20–22], and it is unclear whether Roquin proteins recruit the CCR4-NOT complex in invertebrates. However, the conservation of the ROQ domain indicates that Roquin proteins also bind RNA in invertebrates, although their specific RNA targets are currently unknown.

Despite extensive information regarding the mode of RNA recognition by Roquin proteins[15–19], a detailed molecular understanding of how the proteins interact with the CCR4-NOT complex is lacking, and it is not even known whether the interactions are direct. Here we investigate the molecular details of how Roquin proteins recruit the CCR4-NOT complex. First, we show that Hs Roquin2 and Dm Roquin interact with the CCR4-NOT complex and promote target mRNA degradation via their C-terminal regions, suggesting conserved functional principles among all Roquin proteins. Furthermore, we find that the Dm Roquin C-terminal region contains multiple binding sites for the CCR4-NOT complex and that these sites act redundantly to promote mRNA degradation. Among these sites, we identify a short linear motif (SLiM) that is necessary and sufficient to mediate direct binding to the CAF40 subunit of the CCR4-NOT complex. This motif is termed the CAF40-binding motif (CBM), and we determine its crystal structure bound to CAF40. Structure-based mutations of the CAF40-CBM interface prevent binding of Dm Roquin to CAF40 and reduce the ability of the protein to degrade mRNA targets, indicating that CAF40 is an important mediator of the recruitment of the CCR4-NOT complex. Together with previous studies[10], our results reveal a common role of the Roquin C-terminal region as an effector domain that regulates mRNA target expression through the recruitment of the CCR4-NOT complex despite the lack of sequence conservation.

## Results

**Roquin C-terminal regions recruit the CCR4-NOT complex.** Hs Roquin1 interacts with the CCR4-NOT deadenylase complex through a C-terminal region that shows very low sequence similarity to the C-terminal regions of the corresponding vertebrate Roquin2 paralogues and invertebrate Roquin[10] (Fig. 1a). We therefore asked whether the C-terminal regions of Hs Roquin2 and Dm Roquin have the ability to interact with the CCR4-NOT complex. Hs Roquin2 and Dm Roquin expressed with a tag consisting of the V5 epitope followed by the streptavidin-binding peptide (V5-SBP) pulled down the endogenous CCR4-NOT complex in human HEK293T cells to a similar extent as Hs Roquin1 (Fig. 1b). Moreover, the C-terminal regions of Hs Roquin2 and Dm Roquin were necessary and sufficient for the interaction, as observed for Hs Roquin1 (Fig. 1c,d; Supplementary Fig. 1a)[10]. The observation that Dm Roquin interacts with the CCR4-NOT complex in human cells further suggests that the protein recognizes surfaces on the CCR4-NOT complex that are conserved across species.

**Roquin C-terminal regions mediate mRNA degradation.** We next investigated whether the C-terminal regions of Hs Roquin2 and Dm Roquin elicit the degradation of mRNA targets. To this end, we used an MS2-based tethering assay in human HEK293T cells. The full-length proteins and the corresponding N- and C-terminal fragments (Hs Roq2-N, Hs Roq2-C, Dm Roq-N and Dm Roq-C, respectively; Fig. 1a, Supplementary Table 1) were expressed with an MS2-HA tag that mediates binding to a β-globin reporter mRNA containing six MS2-binding sites in its 3′-UTR (β-globin-6xMS2bs)[33]. Tethered Hs Roquin2 and Dm

Roquin reduced the level of the β-globin-6xMS2bs mRNA relative to the MS2-HA fusion protein, which was used as a negative control (Fig. 1e–j). Furthermore, the C-terminal fragments retained the mRNA degradation activity of the full-length proteins, whereas the N-terminal fragments were inactive (Fig. 1e–j). Similar results were obtained for *Hs* Roquin1 (Supplementary Fig. 1b–d). The N- and C-terminal fragments were expressed at levels comparable to those of the full-length

proteins (Fig. 1g,j), and none of the proteins affected the expression of the control β-globin mRNA lacking MS2-binding sites (Fig. 1f,i; control). Thus we conclude that, despite the lack of sequence conservation, the C-terminal regions of Roquin proteins interact with the CCR4-NOT complex and promote the degradation of bound mRNAs.

As shown for *Hs* Roquin1 and Roquin2, *Dm* Roquin regulates the expression of a β-globin mRNA reporter containing the

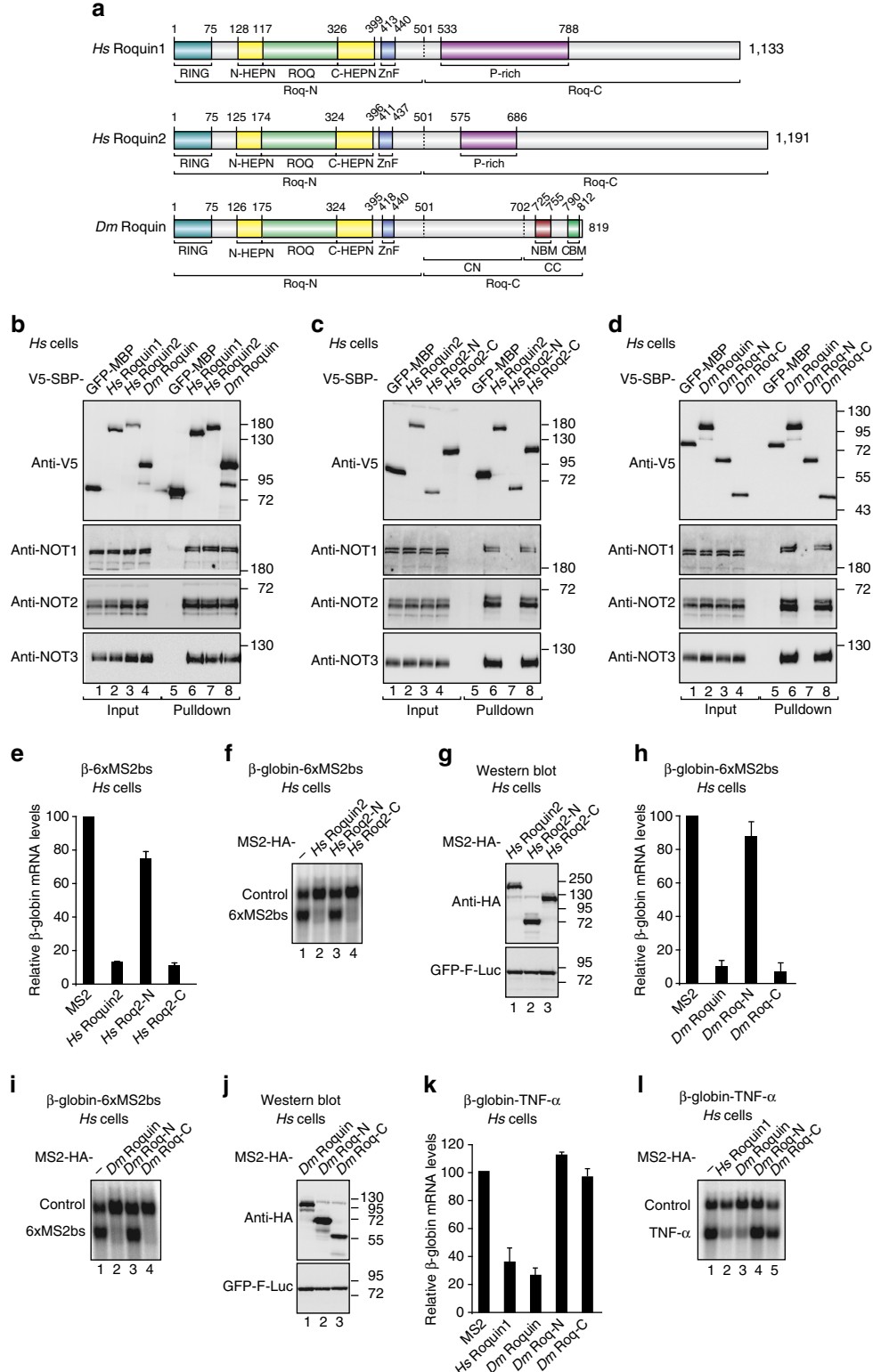

constitutive decay element (CDE) from the TNF-α mRNA in its 3′-UTR (CDE-37; ref. 10) in human HEK293T cells (Fig. 1k,l), consistent with the conservation of the ROQ domain[15,16,23]. Regulation of the β-globin-TNF-α reporter by *Dm* Roquin was abolished by mutations in the CDE that disrupt the binding of the ROQ domain (Supplementary Fig. 1e,f; mutant MUT16; ref. 10). Furthermore, in contrast to the results obtained in the tethering assay, degradation of the β-globin-TNF-α reporter required both the N- and the C-terminal regions of *Dm* Roquin (Fig. 1k,l). Indeed, the C-terminal region alone was not sufficient to cause degradation of the β-globin-TNF-α reporter (Fig. 1k,l), most likely because it does not bind RNA.

**Roquin proteins direct mRNAs to the 5′-to-3′ decay pathway.** Given that Roquin proteins interact with the CCR4-NOT complex, we next investigated whether the proteins elicit degradation of mRNA targets via the 5′-to-3′ decay pathway, in which deadenylation is followed by decapping and 5′-to-3′ exonucleolytic degradation of the mRNA body. To this end, we performed tethering assays in HEK293T cells overexpressing a catalytically inactive DCP2 mutant (DCP2 E148Q), which inhibits decapping in a dominant-negative manner[34]. We observed that degradation of the β-globin-6xMS2bs mRNA by tethered *Hs* Roquin1, *Hs* Roquin2 and *Dm* Roquin was impaired in cells expressing the DCP2 mutant; in these cells, the reporter accumulated in a shorter deadenylated form (Fig. 2a,b, lanes 6–8). The expression of the tethered proteins was not affected by the expression of DCP2 mutant (Fig. 2c). Our results indicate that the three Roquin proteins direct mRNA targets to the 5′-to-3′ decay pathway.

**Dm Roquin degrades bound mRNAs in Drosophila cells.** To investigate whether the *Dm* Roquin protein has the ability to repress and degrade bound mRNAs in *Dm* cells, we used a λN-based tethering assay in *Drosophila melanogaster* Schneider S2 cells[35]. Tethered *Dm* Roquin caused 10-fold repression of a firefly luciferase (F-Luc) reporter containing five binding sites for the λN-tag (BoxB hairpins) in its 3′-UTR (Fig. 3a–c). The reduction in F-Luc activity was accompanied by a corresponding decrease in mRNA abundance (Fig. 3a–c) and a decrease in the half-life of the mRNA (Fig. 3d,e), indicating that *Dm* Roquin induces mRNA degradation in S2 cells. *Dm* Roquin did not affect the expression of an F-Luc reporter that lacked the BoxB hairpins (Supplementary Fig. 1g,h).

As observed in human cells, the Roq-C fragment retained the repressive activity of the full-length protein in the tethering assay and accelerated the degradation of the mRNA reporter (Fig. 3a–e), whereas the Roq-N fragment was inactive (Fig. 3a,b).

Furthermore, the full-length *Dm* Roquin and the Roq-C fragment repressed the translation of a F-Luc mRNA reporter with a 3′-end generated by a self-cleaving hammerhead ribozyme (F-Luc-5BoxB-$A_{95}C_7$-HhR; Fig. 3f,g). This reporter is resistant to deadenylation and is therefore not degraded in S2 cells[36]. Similar results have been reported for other RNA-associated proteins that recruit the CCR4-NOT complex[8,9,36], although the involvement of other factors cannot be excluded.

To confirm that mRNA degradation caused by *Dm* Roquin is dependent on the CCR4-NOT complex, we depleted NOT1 in S2 cells. The ability of *Dm* Roquin to elicit the degradation of the F-Luc-5BoxB mRNA was partially suppressed in NOT1-depleted cells (Fig. 3h,i). Western blotting analysis indicated that the NOT1 levels were reduced to <25% of their control levels in depleted cells (Supplementary Fig. 1i). Thus *Dm* Roquin promotes mRNA degradation by recruiting the CCR4-NOT complex in *Drosophila* cells.

**Dm Roquin interacts directly with CAF40 and the NOT module.** To identify the subunits of the CCR4-NOT complex that interact with *Dm* Roquin, we expressed the GFP-tagged protein in S2 cells and determined whether it interacts with HA-tagged sub-units of the CCR4-NOT complex using co-immunoprecipitation assays. We also tested for interactions of the protein with the PAN2-PAN3 deadenylase complex and with decapping factors. *Dm* Roquin interacted with NOT1, NOT2, NOT3, CAF40, NOT10, CAF1 and PAN3 (Supplementary Fig. 2a–j). These interactions were observed in the presence of RNase A, suggesting that they are not mediated by RNA. *Dm* Roquin also interacted with the decapping factor HPat in an RNA-independent manner but not with other decapping factors (Supplementary Fig. 2k–o). In particular, and in contrast to *Hs* Roquin1 (ref. 32), we observe no interaction of *Dm* Roquin with *Dm* EDC4 or Me31B, which is the *Dm* orthologue of DDX6/RCK (Supplementary Fig. 2n,o).

To discriminate between direct and indirect interactions with subunits of the CCR4-NOT complex, we performed pulldown assays *in vitro* using purified recombinant proteins expressed in *Escherichia coli*. Because *Dm* NOT1 is not expressed in a soluble form in bacteria, we expressed the human proteins, which interact with *Dm* Roquin (as shown in Fig. 1b) and tested them for interaction with *Dm* Roquin *in vitro*.

Initially, we used a purified human pentameric complex consisting of a NOT1 fragment comprising residues 1093–2376, CAF1, CAF40 and the C-terminal domains of NOT2 and NOT3 (Fig. 4a). The *Dm* Roq-C fragment carrying an N-terminal maltose-binding protein (MBP) tag pulled down the purified pentameric complex (Fig. 4b, lane 20), thus demonstrating its direct interaction with the complex.

**Figure 1 | The C-terminal regions of Roquin proteins interact with the CCR4-NOT complex and induce degradation of bound mRNA.** (**a**) Roquin proteins consist of a conserved N-terminal region containing a RING-finger E3 ubiquitin ligase domain, a ROQ RNA-binding domain flanked by a bilobed HEPN domain and a CCCH-type zinc-finger (ZnF) domain. The N-terminal region is followed by a variable C-terminal extension (shown in grey) that often contains proline-rich sequences (P-rich). The positions of the CBM and the NBM in *Dm* Roquin are indicated. The numbers above the protein outline indicate the residues at domain/motif boundaries. (**b–d**) SBP pulldown assays in HEK293T cells expressing V5-SBP-tagged *Hs* Roquin1, *Hs* Roquin2 and *Dm* Roquin (full-length or N- and C-terminal fragments). A V5-SBP-tagged GFP-MBP fusion served as a negative control. The presence of endogenous NOT1, NOT2 and NOT3 in the bound fractions was analysed by western blotting using specific antibodies. The inputs (1.5% for the V5-SBP tagged proteins and 1% for NOT1, NOT2 and NOT3) and bound fractions (10% for the V5-SBP tagged proteins and 30% for NOT1, NOT2 and NOT3) were analysed by western blotting. (**e–j**) Tethering assays using the β-globin-6xMS2bs reporter and MS2-HA-tagged *Hs* Roquin2 and *Dm* Roquin (full-length or the indicated fragments) in human HEK293T cells. A plasmid expressing a β-globin mRNA reporter lacking MS2-binding sites (Control) served as a transfection control. The β-globin-6xMS2bs mRNA level was normalized to that of the control mRNA and set to 100% in cells expressing MS2-HA. The mean values ± s.d. from three independent experiments are shown in **e,h**. (**f,i**) show representative northern blottings. (**g,j**) show the equivalent expression of the MS2-HA-tagged proteins used in the corresponding tethering assays. (**k,l**) Effect of *Dm* Roquin on the expression of the β-globin-TNF-α mRNA reporter analysed as described in **e–j**. Protein size markers (kDa) are shown on the right of the western blotting panels. Error bars represent s.d. from three independent experiments. Full images of western and northern blottings are shown in Supplementary Fig. 10.

To map the binding site more precisely, we tested interactions with individual CCR4-NOT subcomplexes, including the NOT1 MIF4G domain bound to CAF1, the NOT1 CN9BD bound to CAF40, a C-terminal connector domain of unknown function (CD) and the NOT module (comprising the NOT1 SHD and the C-terminal regions of NOT2 and NOT3). MBP-tagged *Dm*

Roq-C pulled down the CN9BD-CAF40 complex as well as the NOT module but not the NOT1 MIF4G-CAF1 complex or the CD (Fig. 4b, lanes 21–24).

Further analysis indicated that *Dm* Roq-C interacted directly with both *Hs* and *Dm* CAF40 in the absence of the NOT1 CN9BD (Fig. 4c, lanes 11 and 12). We also investigated whether the interaction of *Dm* Roq-C with the NOT module was mediated by the NOT1 C-terminal SHD domain or by the NOT2-NOT3 dimer. However, splitting the NOT module resulted in severely reduced binding to both NOT1 and the NOT2-NOT3 dimer, demonstrating that only the assembled module is recognized efficiently (Supplementary Fig. 3a). In summary, the *Dm* Roq-C fragment contains at least two distinct binding sites for the CCR4-NOT complex, one site that contacts CAF40 and a second site that contacts the NOT module.

**Redundancy of CCR4-NOT-binding motifs in *Dm* Roq-C.** To define more precisely how *Dm* Roq-C interacts with CAF40 and the NOT module, we sought to identify conserved motifs within the primary sequence. Using only sequences from *Drosophila* species, it was possible to align the *Dm* Roq-C sequences across their entire length (Supplementary Fig. 4). The alignment revealed clusters of conserved residues dispersed throughout the sequence with a higher level of conservation evident at the C-terminal end of Roq-C (Supplementary Fig. 4). We therefore generated two fragments, which we termed Roq-CN and Roq-CC (Fig. 1a, Supplementary Fig. 4). Remarkably, each of these fragments in isolation exhibited repressive activity in tethering assays, indicating functional redundancy (Supplementary Fig. 3b–h). However, only the Roq-CC fragment, comprising residues 702–819, bound to the purified CAF40 protein *in vitro* as efficiently as the entire Roq-C fragment, whereas binding of the Roq-CN fragment (residues 501–702) was strongly impaired (Fig. 4d, lanes 23 and 20 versus lane 17). In contrast, both Roq-CN and Roq-CC retained binding activity for the NOT module, although their binding was reduced compared with that of the full Roq-C fragment (Fig. 4d, lanes 21 and 24 versus lane 18). These results indicate the presence of multiple binding sites for the NOT module within Roq-C.

Through a deletion analysis combined with binding, we then identified a motif comprising residues 790–812 within Roq-CC that is necessary and sufficient for binding to the CAF40 armadillo repeat (ARM) domain (Figs 1a and 4e). Indeed, deletion of the CBM in Roq-CC abolished its binding to CAF40 (Figs 4e, lane 14). Conversely, the CBM is sufficient for binding to CAF40 (Fig. 4e, lane 16 versus lane 12). We also identified

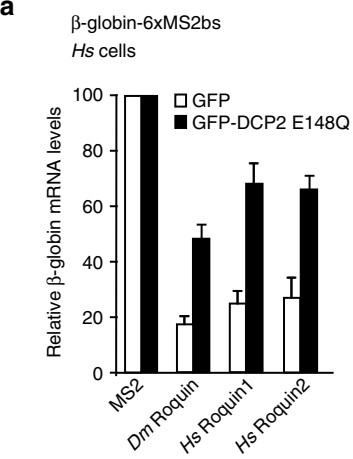

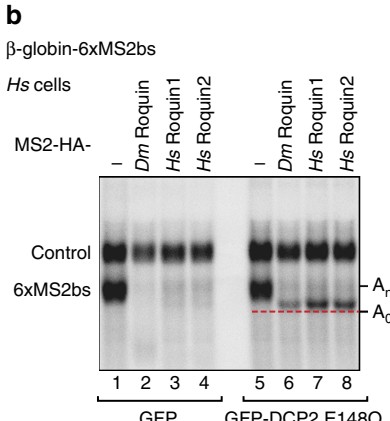

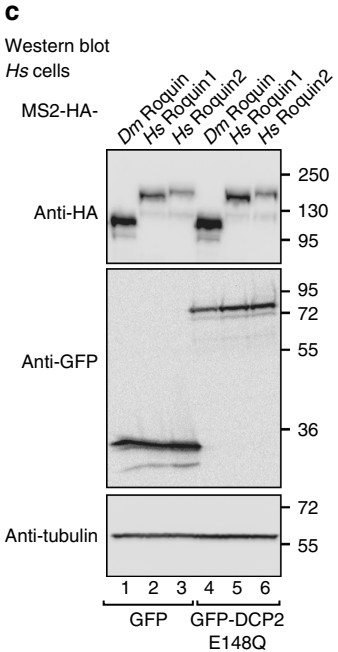

**Figure 2 | Roquin proteins degrade mRNAs through the 5′-to-3′ mRNA decay pathway.** (**a,b**) A tethering assay was performed in HEK293T cells using the β-globin-6xMS2bs reporter as described in Fig. 1e–j, except that different amounts of plasmid were transfected (see Methods section). The transfection mixture included plasmids expressing either GFP or a GFP-tagged catalytically inactive DCP2 mutant (E148Q). The β-globin-6xMS2bs mRNA levels were normalized to those of the control mRNA and set to 100% in the presence of MS2-HA for each condition. The mean values ± s.d. from three independent experiments are shown in **a**. The white and black bars represent the normalized β-globin-6xMS2bs mRNA values in cells expressing GFP and the GFP-DCP2 mutant, respectively. (**b**) shows a representative northern blotting. The positions of the polyadenylated ($A_n$) and deadenylated ($A_0$, dashed red line) mRNA reporter are indicated on the right. (**c**) Western blotting analysis showing equivalent expression of MS2-HA-tagged proteins in cells expressing GFP or the DCP2 mutant. Error bars represent s.d. from three independent experiments. Full images of western and northern blottings are shown in Supplementary Fig. 11.

residues 725–755 as being required for binding to the NOT module in the context of Roq-CC (this region was termed the NOT-module binding motif (NBM); Fig. 1a, Supplementary

Fig. 5a, lane 14); however, in isolation the NBM was not sufficient for binding, indicating a more complicated binding mode (Supplementary Fig. 5a, lane 16).

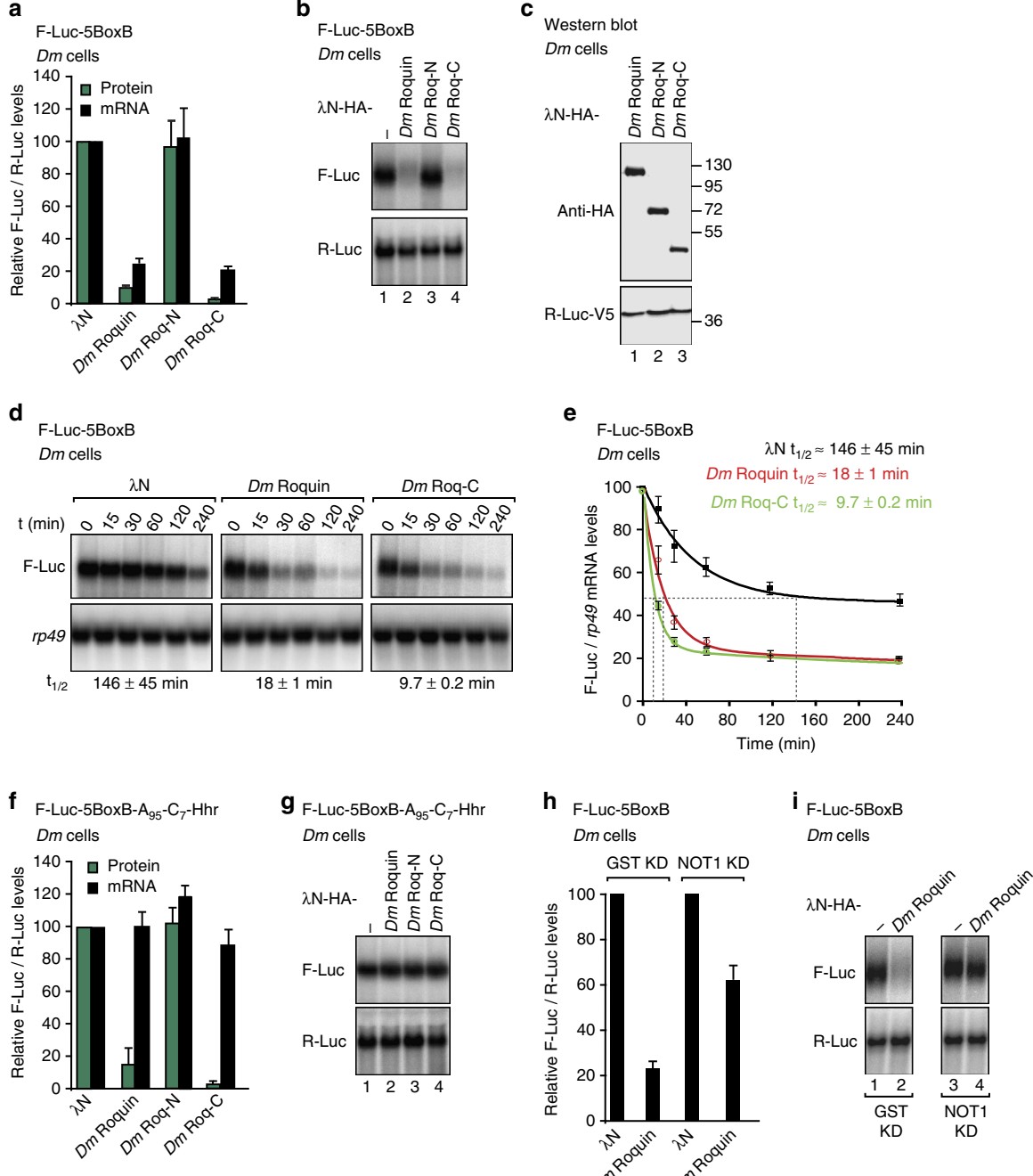

**Figure 3 | *Dm* Roquin degrades bound mRNAs.** (**a**,**b**) Results of tethering assays using the F-Luc-5BoxB reporter and λN-HA-tagged *Dm* Roquin (full-length or the indicated fragments) in *Dm* S2 cells. A plasmid expressing R-Luc served as a transfection control. F-Luc activity and mRNA levels were normalized to those of the R-Luc transfection control and set to 100% in cells expressing the λN-HA peptide. The mean values ± s.d. from three independent experiments are shown in **a**. (**b**) shows a representative northern blotting. The corresponding control experiment with a F-Luc reporter lacking the BoxB sites is shown in Supplementary Fig. 1g,h. (**c**) Western blotting showing the equivalent expression of the λN-HA-tagged proteins used in **a**,**b**. (**d**,**e**) Representative northern blotting showing the decay of the F-Luc-5BoxB mRNA in S2 cells expressing λN-HA or λN-HA-tagged *Dm* Roquin or the Roq-C fragment after inhibition of transcription by actinomycin D. F-Luc mRNA levels were normalized to those of the *rp49* mRNA and plotted against time. The mRNA half-life ($t_{1/2}$) ± s.d. was calculated from the decay curve shown in **e**. (**f**,**g**) Results of the tethering assays using the F-Luc-5BoxB-A$_{95}$-C$_7$-HhR reporter and λN-HA-tagged *Dm* Roquin (full-length or the indicated fragments) in *Dm* S2 cells. The samples were analysed as described in **a**,**b**. (**h**,**i**) Tethering assay using the F-Luc-5BoxB reporter and λN-HA-tagged *Dm* Roquin in *Dm* S2 cells depleted of NOT1 or in control cells (treated with a dsRNA targeting GST). The samples were analysed as described in **a**,**b**. The efficacy of the depletion is shown in Supplementary Fig. 1i. Error bars represent s.d. from three independent experiments. Full images of western and northern blottings are shown in Supplementary Fig. 12.

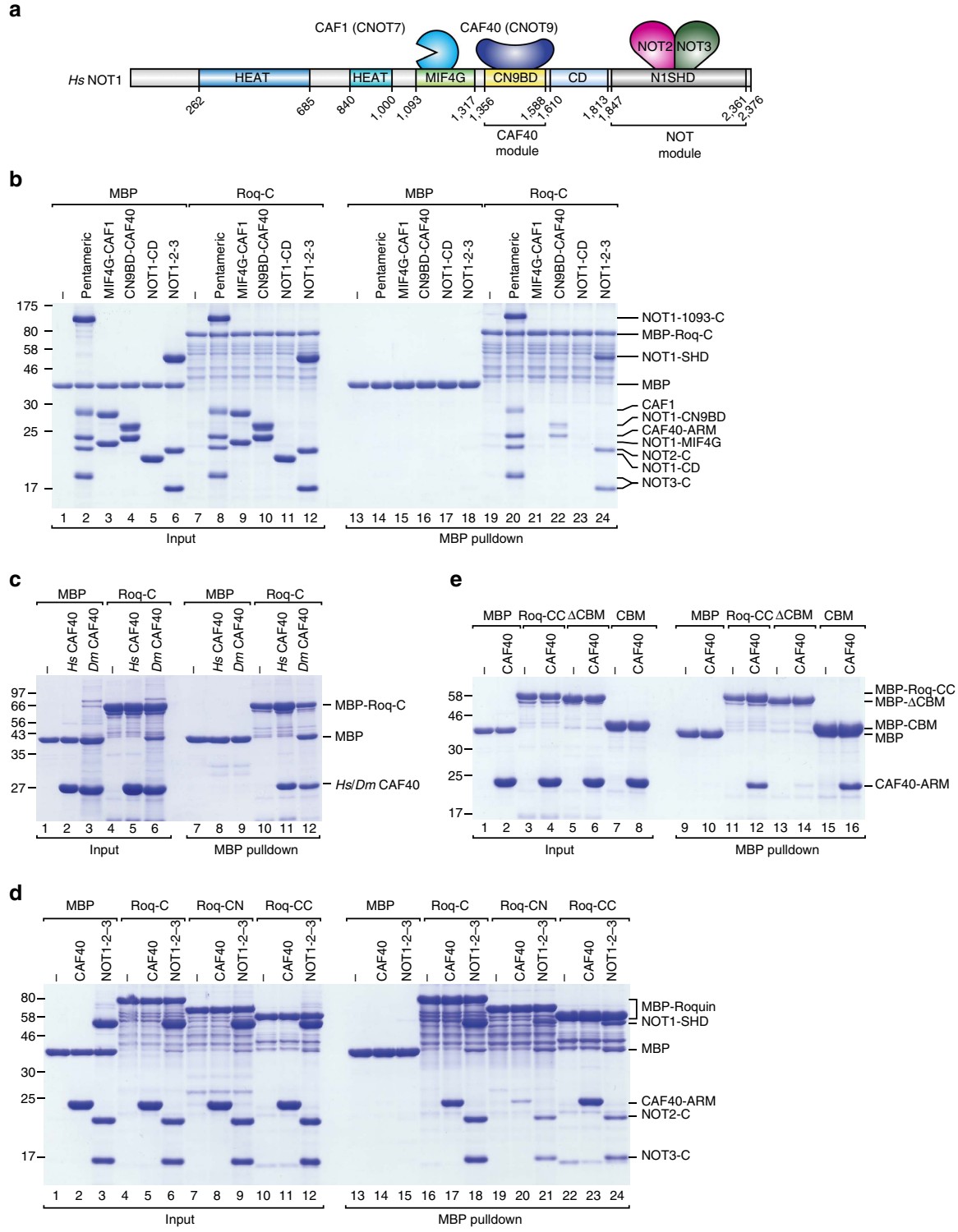

**Figure 4 | *Dm* Roq-C interacts directly with CAF40 and the NOT module.** (**a**) Schematic representation of the *Hs* CCR4-NOT complex with the subunits used in this study. NOT1 contains two HEAT repeat domains (shown in blue and petrol), a MIF4G domain composed of HEAT repeats (shown in green), a three-helix bundle domain (CN9BD, yellow), a connector domain (CD, light blue) and a NOT1 superfamily homology domain (SHD, grey), which also consists of HEAT repeats. The positions of the other subunits indicate their binding sites on NOT1. (**b**) MBP pulldown assay showing the interaction of MBP-tagged *Dm* Roq-C with purified pentameric NOT1-2-3-7-9 complex, the indicated NOT1 domains and CCR4-NOT subcomplexes. MBP served as a negative control. The difference in the migration of the NOT3-C protein in different samples is due to the presence of a non-cleavable 6xHis-tag in the case of the pentameric NOT1-2-3-7-9 complex, whereas in the case of the NOT1-2-3 complex the HRV3C-cleavable 6xHis tag was removed during purification. (**c**) MBP pulldown assay showing the interaction of MBP-tagged *Dm* Roq-C with purified *Hs* and *Dm* CAF40 proteins. (**d**) MBP pulldown assay showing the interaction of MBP-tagged *Dm* Roquin fragments (Roq-C, Roq-CN and Roq-CC) with the *Hs* CAF40 proteins and the recombinant *Hs* NOT module (NOT1-2-3) containing the NOT1 SHD and the NOT2 and NOT3 C-terminal fragments. (**e**) MBP pulldown assay showing the interaction of MBP-tagged *Dm* Roquin fragments (Roq-CC, Roq-CC-ΔCBM or CBM alone) with the *Hs* CAF40-ARM domain. Protein size markers (kDa) are shown on the left in each panel. Full images of protein gels are shown in Supplementary Fig. 13.

We next tested whether the interaction with CAF40 is also observed for the human Roquin proteins. We observed that *Hs* CAF40 interacts directly with the C-terminal region of

*Hs* Roquin1 (Supplementary Fig. 5b); however, more detailed mapping to identify a single CBM was unsuccessful, thus suggesting the presence of multiple binding sites.

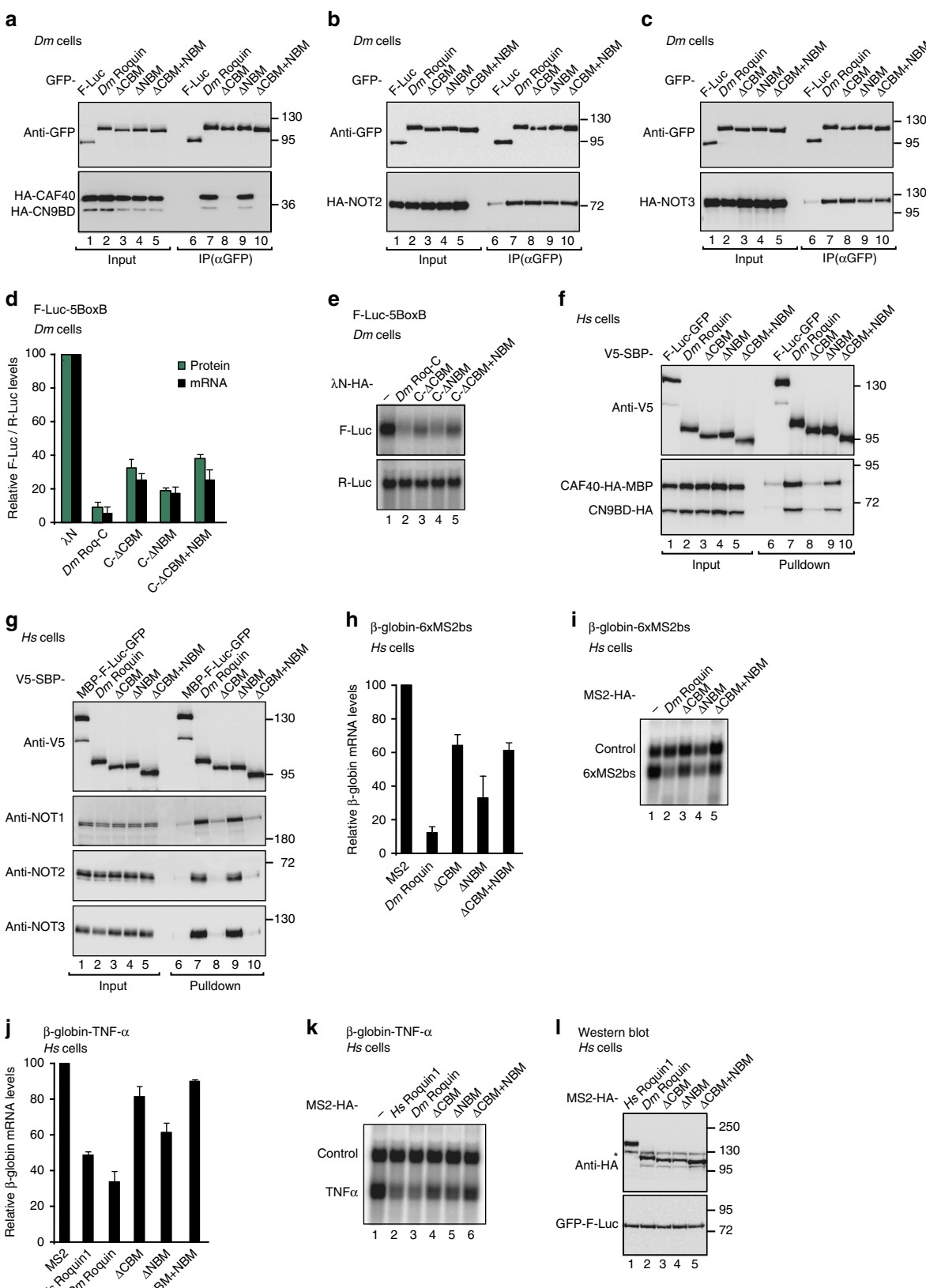

**The CBM contributes to the activity of *Dm* Roquin**. To assess the relative contribution of the CBM and the NBM to the binding of *Dm* Roquin to the CCR4-NOT complex and to the repressive activity of *Dm* Roquin, we deleted these motifs from the protein and performed co-immunoprecipitations and tethering assays in *Dm* S2 cells and human HEK293T cells. Deletion of the CBM abolished the interaction of full-length *Dm* Roquin with the *Dm* CAF40-CN9BD module in S2 cells (Fig. 5a, lane 8). However, this deletion did not affect the binding of *Dm* Roquin to NOT2 or NOT3 (Fig. 5b,c, lane 8). Deletion of the NBM had no effect on CAF40, NOT2 or NOT3 binding (Fig. 5a–c, lane 9), consistent with the observation that *Dm* Roquin harbours multiple binding

sites for the NOT module. Accordingly, deletion of the CBM and NBM in the context of Roq-C reduced but did not abolish the activity of this fragment in tethering assays (Fig. 5d,e, Supplementary Fig. 5c–e).

Similarly, in human cells, deletion of the CBM abolished the interaction of full-length *Dm* Roquin with the CAF40-CN9BD module (Fig. 5f, lane 8), indicating that the CBM represents the only binding site for CAF40 in *Dm* Roquin. Importantly, deletion of the CBM also abolished the interaction of full-length *Dm* Roquin with the endogenous CCR4-NOT complex in human cells (Fig. 5g, lane 8). In agreement with these results, deletion of the CBM reduced the ability of *Dm* Roquin to degrade β-globin-6xMS2bs and β-globin-TNF-α mRNAs (Fig. 5h–l). Furthermore, depletion of CAF40 partially suppressed the activity of *Hs* Roquin1, *Hs* Roquin2 and *Dm* Roquin in tethering assays (Supplementary Fig. 6a–c), indicating that CAF40 is indeed an important recruitment factor, but other, redundant interactions compensate for the lack of CAF40 in *Dm* and human cells.

In summary, the interaction of the CBM with CAF40 contributes to the recruitment of the CCR4-NOT complex by *Dm* Roquin. However, the observation that deletion of the NBM and CBM have little effect on the activity of full-length *Dm* Roquin in S2 cells probably indicates that *Dm* Roquin establishes additional uncharacterized interactions with the CCR4-NOT complex or additional binding partners to regulate mRNA targets.

**Crystal structure of the *Dm* Roquin CBM bound to CAF40**. To elucidate the molecular principles underlying the recruitment of CAF40 by *Dm* Roquin, we sought to determine the crystal structure of the CBM peptide bound to the *Dm* and *Hs* CAF40 ARM domains, which exhibit 81% identity (Supplementary Fig. 7). However, only the complex containing *Hs* CAF40 yielded well-diffracting crystals, from which we obtained a structure at 2.15 Å resolution (Table 1). Two copies of the complex, which are structurally highly similar, are present in the crystal asymmetric unit (Supplementary Fig. 8a,b; root-mean-square deviation (r.m.s.d.) 0.24 Å over 254 $C_\alpha$ atoms).

As previously described, the CAF40 ARM domain consists of 17 α-helices arranged into six armadillo (ARM) repeats. These repeats adopt the typical crescent-like shape of ARM domains (Fig. 6a–c)[2,3,37], with the concave surface that accommodates the CBM peptide (residues 790–810). The CBM peptide folds into an amphipathic helix (residues 795–810) that runs centrally across the concave surface of CAF40 and binds to a conserved hydrophobic patch close to the previously postulated nucleic acid-binding groove (Fig. 6d–f)[37].

### Table 1 | Data collection and refinement statistics.

| | CAF40–*Dm*CBM |
|---|---|
| *Data collection* | |
| Space group | P2$_1$ |
| No. of reflections | 34,975 |
| Cell dimensions | |
| *a, b, c* (Å) | 56.9, 103.7, 60.8 |
| *α, β, γ* (°) | 90.0, 113.0, 90.0 |
| Wavelength (Å) | 1.00001 |
| Resolution (Å) | 46.8–2.15 (2.20–2.15)* |
| $R_{sym}$ | 0.048 (0.52) |
| $I/\sigma I$ | 13.6 (2.2) |
| Completeness (%) | 98.7 (98.8) |
| Redundancy | 2.9 (2.7) |
| | |
| *Refinement* | |
| Resolution (Å) | 46.7–2.15 |
| No. of reflections | 34,964 |
| $R_{work}/R_{free}$ | 18.4%/22.6% |
| No. of atoms | 4,905 |
| Protein | 4,678 |
| Water | 191 |
| Other solvent molecules | 36 |
| B-factors (Å$^2$) | 60.0 |
| Protein | 60.0 |
| Water | 52.4 |
| Other solvent molecules | 102.5 |
| Ramachandran plot | |
| Favoured regions (%) | 98.8 |
| Disallowed regions (%) | 0.0 |
| r.m.s. deviations | |
| Bond lengths (Å) | 0.004 |
| Bond angles (°) | 0.597 |

*Values in parentheses are for the highest-resolution shell.

**Figure 5 | The CBM contributes to the mRNA degradation activity of *Dm* Roquin.** (**a–c**) Immunoprecipitation assays showing the interaction of GFP-tagged *Dm* Roquin (wild-type or the indicated deletion mutants) with HA-tagged CAF40, NOT2 and NOT3 in *Dm* S2 cells. In (**a**) the interaction was tested in the presence of HA-tagged CN9BD. GFP-tagged firefly luciferase (F-Luc) served as a negative control. Input and immunoprecipitates were analysed using anti-GFP and anti-HA antibodies. For the GFP-tagged proteins, 3% of the input and 10% of the immunoprecipitates were loaded. For the HA-tagged proteins, 1% of the input and 30% of the immunoprecipitates were analysed. In all panels, the cell lysates were treated with RNase A prior to immunoprecipitation. (**d,e**) Tethering assay using the F-Luc-5BoxB reporter and λN-HA-tagged *Dm* Roq-C or the indicated deletion mutants in *Dm* S2 cells. The samples were analysed as described in Fig. 3a,b. The corresponding control experiment with a F-Luc reporter lacking the BoxB sites and a western blotting showing the equivalent expression of the tethered proteins are shown in Supplementary Fig. 5c–e. (**f,g**) Interaction of V5-SBP-tagged *Dm* Roquin (full-length or the indicated deletion mutants) with HA-tagged CAF40 (in the presence of the HA-tagged CN9BD) and with endogenous NOT1, NOT2 and NOT3 in HEK293T cells. A V5-SBP-tagged MBP-F-Luc-GFP fusion served as a negative control. The inputs (0.75% for V5-SBP-tagged proteins and 1% for NOT1, 2, 3) and bound fractions (5% for SBP-V5-tagged proteins and 30% for NOT1, NOT2 and NOT3) were analysed by western blotting. (**h,i**) Tethering assay using the β-globin-6xMS2bs reporter and the indicated MS2-HA-tagged proteins in HEK293T cells. The samples were analysed as described in Fig. 1e–j. (**j,k**) The effect of full-length *Dm* Roquin or the indicated deletion mutants on the expression of the β-globin-TNF-α mRNA reporter was analysed as described in Fig. 1k,l. (**l**) Western blotting analysis showing comparable expression of the MS2-HA-tagged proteins used in **h–k**. The asterisk indicates cross-reactivity with the anti-HA antibody. Error bars represent s.d. from three independent experiments. Full images of western and northern blottings are shown in Supplementary Fig. 14.

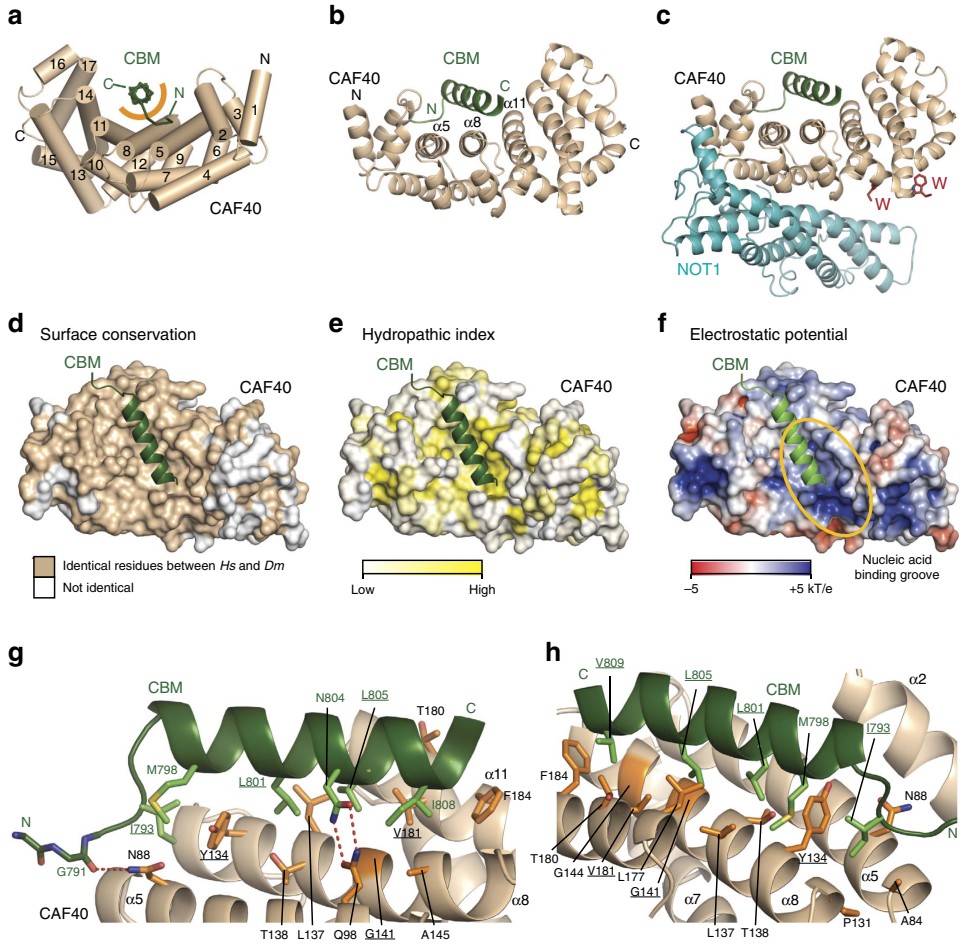

**Figure 6 | Structure of the *Dm* Roquin CBM bound to CAF40.** (**a**) The *Dm* CBM peptide (green; backbone shown in ribbon representation) bound to *Hs* CAF40 (light brown). The helices of CAF40 are depicted as tubes and are numbered in black. The orange semicircle marks the predominantly hydrophobic interface between the CBM peptide and CAF40. (**b**) Cartoon representation of the *Dm* CBM peptide bound to *Hs* CAF40. Secondary structure elements of CAF40 are labelled in black. (**c**) Structural model of the CBM peptide bound to the CAF40 module (consisting of CAF40 bound to the NOT1 CN9BD domain). The model was obtained by superimposing the structure shown in (**b**) with the structure of CAF40 bound to the NOT1 CN9BD domain and free tryptophan (PDB 4CRU)[2]. (**d**) Conservation of the CBM-binding surface on CAF40. CAF40 is shown in surface representation. Surface residues that are identical between *Hs* and *Dm* are shown in light brown; all other residues are shown in white. (**e**) Surface representation of CAF40 with residues coloured in a gradient from white to yellow with increasing hydophobicity[57]. The CBM is shown in green. (**f**) Surface representation of CAF40 with residues coloured in a gradient from red over white to blue according to the electrostatic potential from $-5$ to $+5\,kTe^{-1}$. The electrostatic potential was calculated using the APBS tools plugin within PyMOL (http://www.pymol.org). The proposed nucleic acid-binding groove[37] is indicated by an orange circle. (**g,h**) Close-up views of the CAF40–CBM-binding interface in two orientations. Selected residues of CAF40 and Roquin are shown as orange and green sticks, respectively. Hydrogen bonds are indicated by red dashed lines. The residues that were mutated in this study are underlined.

Superposition of the CAF40-CBM complex with the structure of the CAF40 dimer[37] (r.m.s.d. of 0.87 Å over 260 Cα atoms; PDB 2FV2) or with that of CAF40 bound to the NOT1 CN9BD (r.m.s.d. of 0.59 Å over 254 Cα atoms, PDB 4CRV)[2] shows that binding of the CBM peptide does not induce major conformational changes in the CAF40 ARM domain (Supplementary Fig. 8c,d). Importantly, binding of the CBM does not interfere with NOT1 binding (Fig. 6c), suggesting that *Dm* Roquin can interact with CAF40 in the context of the CCR4-NOT complex. Finally, binding of the CBM does not block access to the tryptophan-binding pockets on the convex surface of CAF40 that serve as binding sites for the GW182/TNRC6 proteins involved in miRNA-mediated gene silencing (Fig. 6c)[2,3].

The amphipathic helix of the CBM peptide lies almost parallel to helices α5, α8 and α11 and it uses residues M798, L801, L805, I808 and V809 on its hydrophobic side to interact with CAF40 residues Y134, L137, G141, G144 and A145 (helix α8) and L177, T180, V181 and F184 (helix α11; Fig. 6g,h and Supplementary

Fig. 8e,f). Furthermore, I793 anchors the N-terminal extension of the CBM helix between CAF40 residues A84, R130, P131 and Y134, resulting in a total buried surface of 1903 Å² that does not include any water molecules (Fig. 6h). Finally, the CBM peptide is fixed by two hydrogen bonds between N804 and the CAF40 residue Q98 (helix α5) and by a hydrogen bond from CAF40 N88 (helix α5) to the carbonyl oxygen of G791 (Fig. 6g). The CAF40 residues R130 and K148 may have additional roles in anchoring the CBM peptide, but they have distinct orientations in the two copies of the complex.

**CCR4-NOT is recruited via the concave surface of CAF40.** To validate the interfaces determined from the crystal structure, we introduced mutations in CBM and CAF40 and tested them in MBP pulldown assays *in vitro*. Substitution of *Dm* Roquin interface residues L805 and V809 by glutamic acid (mutant M2) and the further introduction of I793E and L801E substitutions to create a quadruple mutant (mutant M4) abolished the interaction

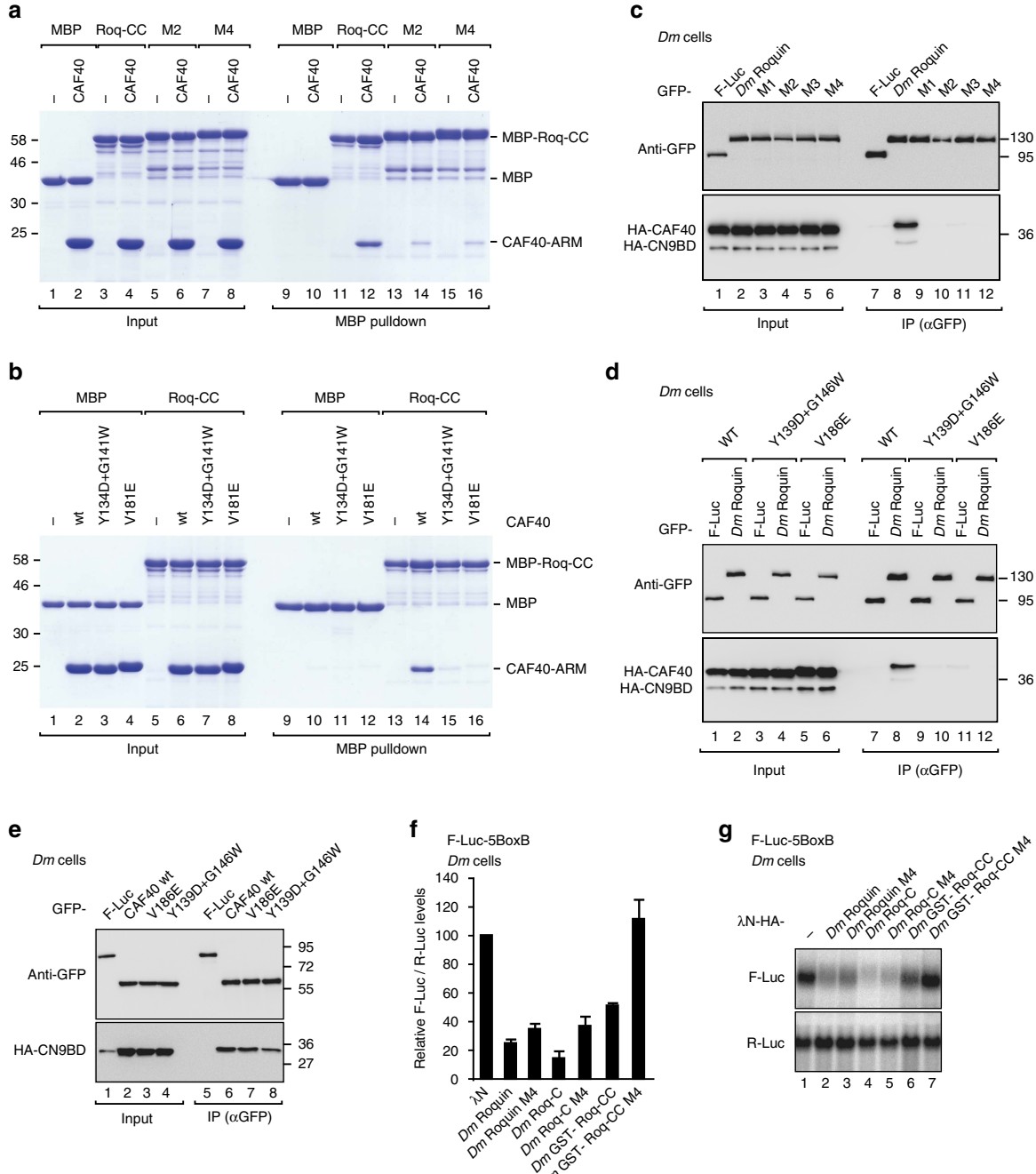

**Figure 7 | The CBM is the only CAF40-binding site in *Dm* Roquin.** (**a**) *In vitro* MBP pulldown assay showing the interaction of MBP-tagged *Dm* Roq-CC and the indicated mutants (M2 and M4; see Supplementary Table 1) with the purified *Hs* CAF40-ARM domain. MBP served as a negative control. (**b**) *In vitro* MBP pulldown assay showing the interaction of MBP-tagged *Dm* Roq-CC with the *Hs* CAF40-ARM domain (wild-type or the indicated mutants). (**c**) Interaction of GFP-tagged *Dm* Roquin (wild-type or the M1, M2, M3 and M4 mutants; see Supplementary Table 1) with HA-tagged CAF40 in the presence of the HA-tagged CN9BD in *Dm* S2 cells. F-Luc-GFP served as a negative control. (**d**) Interaction of GFP-tagged *Dm* Roquin wild-type with HA-tagged CAF40 (wild-type or the indicated mutants) in the presence of the HA-tagged CN9BD in *Dm* S2 cells. (**e**) Interaction of GFP-tagged *Dm* CAF40 (wild-type or mutants) with HA-tagged CN9BD in *Dm* S2 cells. (**f**,**g**) A tethering assay using the F-Luc-5BoxB reporter and λN-HA-tagged *Dm* Roquin, Roq-C and GST-Roq-CC (full-length or mutant M4) was performed in *Dm* S2 cells as described in Fig. 3a,b. (**f**) shows mean values ± s.d. for normalized F-Luc mRNA levels from three independent experiments. (**g**) shows a northern blotting of representative RNA samples corresponding to the experiment shown in **f**. Error bars represent s.d. from three independent experiments. Full images of western and northern blottings are shown in Supplementary Fig. 15.

of the MBP-tagged Roq-CC with purified *Hs* CAF40 *in vitro* (Fig. 7a, Supplementary Table 1). Conversely, a single V181E substitution in *Hs* CAF40 or substitution of residues Y134 and G141 by aspartic acid and tryptophan, respectively, abolished the interaction of *Hs* CAF40 with MBP-Roq-CC (Fig. 7b,

Supplementary Table 1). In contrast, mutation of the concave surface of CAF40 was not sufficient to disrupt the interaction with *Hs* Roq1-C (Supplementary Fig. 5b). Thus, although *Hs* Roq1-C binds to CAF40 directly, it must contact additional and/or alternative CAF40-binding surfaces.

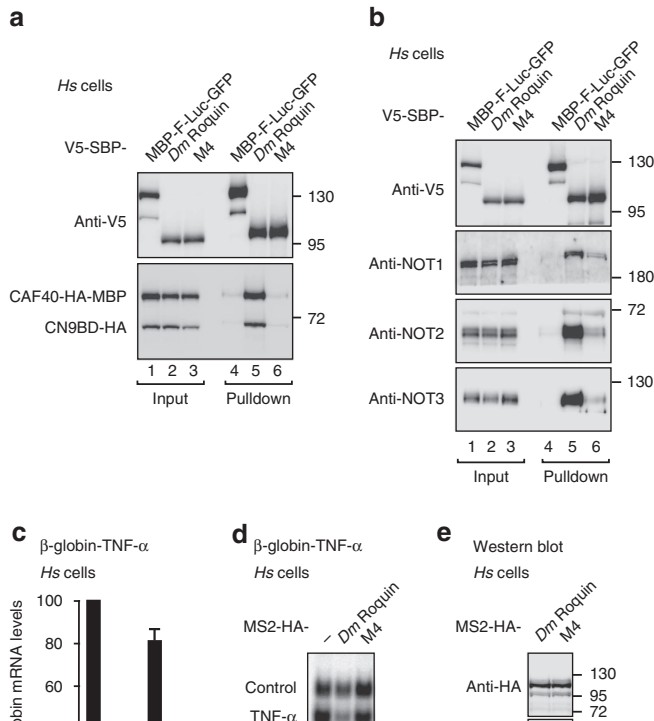

**Figure 8 | The CBM mediates the activity of *Dm* Roquin in human cells.**
(**a**) SBP pulldown assays in HEK293T cell lysates showing the interaction of V5-SBP-tagged *Dm* Roquin (full-length or the M4 mutant) with HA-tagged CAF40-MBP in the presence of CN9BD-HA. A V5-SBP-tagged MBP-F-Luc-GFP fusion served as a negative control. The presence of the HA-tagged proteins in the bound fractions was tested by western blotting using anti-HA antibodies. The samples were analysed by western blotting as described in Fig. 5f. (**b**) SBP pulldown assays in HEK293T cell lysates showing the interaction of V5-SBP-tagged *Dm* Roquin (full-length or the M4 mutant) with endogenous NOT1, NOT2 and NOT3. A V5-SBP-tagged MBP-F-Luc-GFP fusion served as a negative control. The samples were analysed by western blotting as described in Fig. 1b–d. (**c,d**) The effect of *Dm* Roquin full-length or the M4 mutant on the expression of the β-globin-TNF-α mRNA reporter was analysed in HEK293T cells as described in Fig. 1k,l. Error bars represent s.d. from three independent experiments. (**e**) Western blotting analysis showing the equivalent expression of the proteins used in the tethering assays shown in **c,d**. Full images of western and northern blottings are shown in Supplementary Fig. 16.

The residues in CAF40 that interact with the *Dm* Roquin CBM peptide are conserved in *Hs* and *Dm* (Fig. 6d, Supplementary Fig. 7), suggesting that the CBM interacts with *Dm* CAF40 via a similar binding mode. Nevertheless, it was important to test whether *Dm* Roquin binds to the concave surface of *Dm* CAF40. All tested mutations in the CBM (M1, M2, M3 and M4; Supplementary Table 1) were sufficient to disrupt the interaction of full-length *Dm* Roquin with *Dm* CAF40 in co-immunoprecipitation assays in *Dm* S2 cells (Fig. 7c, lanes 9–12), confirming that the CBM is the only motif in *Dm* Roquin that mediates binding to CAF40. Conversely, substitutions in *Dm* CAF40 corresponding to the mutations of *Hs* CAF40 shown in Fig. 7b abolished the interaction of

full-length *Dm* CAF40 with full-length *Dm* Roquin (Fig. 7d). The mutations in CAF40 did not abolish its binding to NOT1 CN9BD, indicating that they do not disrupt the CAF40 fold (Fig. 7e).

Notably, mutations in the CBM did not disrupt the interaction of full-length *Dm* Roquin with NOT2 or NOT3 in *Dm* S2 cells (Supplementary Fig. 9a,b), in agreement with the observation that *Dm* Roquin contains multiple sequences that mediate binding to the CCR4-NOT complex in a redundant manner. Consistent with this redundancy, mutations in the CBM abolished the activity of the Roq-CC fragment in tethering assays in *Dm* S2 cells but impaired the activity of full-length *Dm* Roquin only slightly (Fig. 7f,g, Supplementary Fig. 9c–e). In contrast, when it was tested in human cells, mutations in the *Dm* Roquin CBM not only abolished interaction with CAF40 but strongly reduced the interaction with the endogenous CCR4-NOT complex (Fig. 8a,b) indicating that the CBM provides a major contribution to the interaction of *Dm* Roquin with the CCR4-NOT complex in human cells. Accordingly, the CBM mutants strongly reduced the ability of *Dm* Roquin to degrade the β-globin-TNF-α reporter containing the CDE-37 element in the 3′-UTR in human HEK293T cells (Fig. 8c–e).

## Discussion

In this study, we show that Roquin proteins (*Hs* Roquin1, *Hs* Roquin2, *Dm* Roquin) use their C-terminal extensions to directly recruit the CCR4-NOT complex to mRNA targets, promoting the degradation of these mRNAs. We show that this function is conserved among Roquin proteins despite the fact that the sequences of the unstructured C-terminal regions of these proteins are highly divergent and do not share similar motifs.

In *Dm* Roquin, the interaction with the CCR4-NOT complex is mediated by multiple and partially redundant motifs that include a CBM. We further elucidated the structural basis of the interaction of the CBM with CAF40 and identified the concave surface of CAF40 as a target for amphipathic helices to recruit the CCR4-NOT complex.

The finding that Roquin proteins use a combination of SLiMs (or eukaryotic linear motifs)[38] to recruit the CCR4-NOT complex has important functional implications. First, although SLiMs mediate relatively low-affinity interactions, these interactions can be highly specific, and stable binding can be achieved through avidity effects generated by contributions from the flanking disordered regions that extend the interaction interface[38,39]. In the case of *Dm* Roquin, the sequences flanking the CBM provide binding sites for the NOT module, indicating that the C-terminal region of *Dm* Roquin contacts multiple subunits within the CCR4-NOT complex.

A second consequence of the nature of SLiMs is their evolutionary plasticity[38,39]. Owing to their short length and lack of sequence constraints in the absence of a protein fold, even single point mutations can render an existing motif nonfunctional or generate a new motif in another location of the same protein. In particular, the CBM is present in *Drosophila* species, but sequence analysis of Roquin proteins from other insects, worms and vertebrates does not reveal a detectable CBM. Nevertheless, *Hs* Roquin1 and Roquin2 also interact with the CCR4-NOT complex via their unstructured C-terminal regions (ref. 10 and this study), indicating that the overall principle of CCR4-NOT complex recruitment and mRNA target repression is evolutionarily conserved even though the interaction details have diverged across species.

In addition to the previously identified tryptophan-binding pockets on the CAF40 convex surface[2,3], our crystal structure demonstrates that the concave surface of CAF40 is also used to

recruit the CCR4-NOT complex by RNA-associated proteins. The hydrophobic properties of the concave surface are highly conserved, making it an ideal partner for amphipathic helices or disordered, short hydrophobic peptides that are frequently present in RNA-binding proteins.

Therefore, the C-terminal regions of *Hs* Roquin1 and Roquin2 may also target this surface of CAF40, although no clear CBM motifs are detectable by sequence analysis. However, the proteins may contain 'cryptic' CBMs that are likely discontinuous and probably also target neighbouring surfaces on CAF40 because mutations affecting the concave surface of *Hs* CAF40 did not disrupt its interaction with the *Hs* Roquin1 C-terminal fragment.

In addition to the Roquin proteins, a large number of RNA-associated proteins have been shown to recruit the CCR4-NOT complex to their mRNA targets, thereby repressing translation and/or promoting mRNA degradation. These include GW182 proteins, TTP, Nanos and the *Dm* proteins CUP, Bicaudal C, Smaug and Pumilio[2,3,6,8,9,40–44]. For most of these proteins, it has been shown that interaction with the CCR4-NOT complex is mediated by SLiMs embedded in peptide regions of predicted disorder. However, detailed characterization of the binding mode on the structural level is only available for TTP, GW182, vertebrate and *Dm* Nanos and *Dm* Roquin (refs 2,3,6,8,9 and this study). Similar to Roquin, *Dm* Nanos, GW182 proteins and TTP all contain multiple binding sites for different subunits of the CCR4-NOT complex that act redundantly to recruit the complex to mRNA targets. This modular recruitment mode likely enhances the binding affinity and confers redundancy and robustness to the repression mechanism.

As observed for Roquin, the motifs in TTP and vertebrate and *Dm* Nanos, which have been structurally characterized, adopt an α-helical conformation that possibly forms only upon binding (refs 6,8,9 and this study). In contrast, GW182 peptides likely bind to the CCR4-NOT complex in an extended conformation and insert tryptophan residues into tandem hydrophobic pockets exposed at the convex surface of CAF40 and into additional pockets in NOT1 that remain to be identified[2,3].

Because *Dm* Roquin and GW182 proteins can bind CAF40 simultaneously (Fig. 6c), it is possible that the proteins cooperate to recruit the CCR4-NOT complex to mRNAs. The presence of multiple CCR4-NOT-binding partners on an mRNA likely enhances the efficiency of recruitment and the extent of the regulation.

In summary, together with previous studies[2,3,6,8,9], our results indicate that SLiMs in unstructured and poorly conserved regions of RNA-associated proteins represent a common and widespread mode of recruitment of the CCR4-NOT complex to mRNA targets, resulting in a common downstream repressive mechanism that has a major role in posttranscriptional mRNA regulation in eukaryotic cells.

## Methods

**DNA constructs.** Luciferase reporters and plasmids for the expression of GFP- and HA-tagged subunits of the CCR4-NOT and PAN2-PAN3 deadenylase complexes and decapping factors in S2 cells were previously described[35,45,46]. *Dm* Roquin cDNA was purchased from the *Drosophila* Genomics Resource Center, amplified by PCR and inserted into the HindIII and NotI restriction sites of the pAc5.1-λN-HA or pAc5.1-GFP vectors. For expression in HEK293T cells, the cDNA encoding *Dm* Roquin was inserted into the NotI and ApaI restriction sites of the pcDNA3.1-MS2-HA vector[33] and between the HindIII and KpnI sites of the pT7-V5-SBP-C1 vector[47].

A cDNA sequence encoding *Hs* Roquin1 open-reading frame was amplified by PCR from human HEK293T cell total cDNA and inserted between the EcoRI and SacII sites of the pT7-V5-SBP-C1 vector and between the NotI and ApaI sites of the pcDNA3.1-MS2-HA vector. A cDNA encoding *Hs* Roquin2 open-reading frame was amplified by PCR from human HeLa cell total cDNA and inserted between the BamHI and NotI restriction sites of the pcDNA3.1-MS2-HA vector and between the SalI and BamHI restriction sites of the pT7-V5-SBP-C1 vector.

β-Globin reporters containing the wild-type or mutant (MUT16) TNF-α CDE were obtained by replacing the 6xMS2-binding sites in vector β-globin-6xMS2bs[33] with the CDE-37 (TTGGCTCAGACATGTTTTCCGTGAAAACGGAGCTGAA) or CDE-37-MUT16 (TTGGCTCAGACATGTTTTCCGTGAAATGGGAGCT GAA) sequences[10].

For expression of recombinant proteins in *E.coli*, cDNAs encoding *Hs* Roq1-C and *Dm* Roquin fragments were inserted between the AflII and AvrII and the AflII and XbaI restriction sites of the pnYC-pM plasmid[48], respectively, resulting in Roquin fusion proteins carrying N-terminal MBP tags that are cleavable by HRV3C protease. The Roq-C and Roq-CC cDNAs and all of the constructs derived from them also contain a C-terminal GB1 tag[49] followed by a MGSS linker and a 6xHis tag.

A cDNA encoding *Dm* CAF40 was inserted between the NdeI and XbaI restriction sites of the pnEA-pM plasmid[48], generating a fusion protein containing an N-terminal MBP tag cleavable by the HRV3C protease.

For purification of the pentameric NOT1-2-3-7-9 complex, a cDNA encoding human NOT1 (residues 1093–2371) was inserted between the XhoI and BamHI restriction sites of the pnYC-pM vector, resulting in a fusion protein containing an N-terminal MBP-tag. A multicistronic plasmid was generated by inserting an expression cassette containing 6xHis-NOT3-C, MBP-NOT2-C, 6xHis-CAF40-ARM and GST-NOT7 (all tags except the CNOT3 6xHis tag are cleavable by HRV3C protease) into the pnEA vector.

A cDNA encoding NOT1-CD (residues 1607–1815) was inserted between the XhoI and BamHI restriction sites of the pnYC-pM plasmid[48], generating a fusion protein containing an N-terminal MBP tag cleavable by HRV3C protease.

Plasmids for the expression of NOT1-MIF4G, NOT1-CN9BD, NOT1-SHD, NOT2-C, NOT3-C, NOT7 and CAF40-ARM have been previously described[2,5,50]. The DNA constructs used in this study are listed in Supplementary Table 1.

**Co-immunoprecipitation and SBP-pulldown assays.** For co-immunoprecipitation assays in S2 cells (ATCC), $2.5 \times 10^6$ cells were seeded per well in six-well plates and transfected using Effectene transfection reagent (Qiagen). The transfection mixtures contained 1 μg of plasmid expressing HA-tagged deadenylase or decapping factors and 1.5 μg of GFP-tagged Roquin (either full length or fragments). S2 cells were harvested 3 days after transfection and co-immunoprecipitation assays were performed as previously described[51].

For SBP pulldown assays in human cells, HEK293T cells (ATCC) were grown in 10-cm dishes and transfected using TurboFect transfection reagent (Thermo Fisher Scientific). The transfection mixtures contained 6, 30 and 2 μg of plasmids expressing *Hs* Roquin1, *Hs* Roquin2 and *Dm* Roquin, respectively. In the experiment shown in Fig. 1c, the transfection mixture contained 25 μg of plasmids expressing *Hs* Roquin2 and Roq2-N and 12 μg of a plasmid expressing Roq2-C. In the experiment shown in Fig. 1d, the transfection mixtures contained 15, 20 and 25 μg of plasmids expressing *Dm* Roquin, Roq-N and Roq-C, respectively. In the experiments shown in Fig. 5f,g, 10 μg of plasmids expressing *Dm* Roquin, ΔCBM and ΔNBM and 15 μg of plasmid expressing *Dm* Roquin ΔCBM + NBM were included. In the experiments shown in Fig. 8a,b, 10 μg of the *Dm* Roquin and *Dm* Roquin M4 plasmids was transfected. In the experiments shown in Figs 5f and 8a, the transfection mixtures contained 7.5 μg of a plasmid expressing CAF40-HA-MBP and 5 μg of a plasmid expressing CN9BD-HA. In the experiments shown in Supplementary Fig. 1a, the transfection mixtures contained 1 μg of plasmids expressing *Hs* Roquin1 and Roq1-C and 10 μg of a plasmid expressing Roq1-N. Human cells were harvested 2 days after transfection, and co-immunoprecipitation assays were performed as previously described[8]. Western blottings were developed using the ECL Western Blotting Detection System (GE Healthcare) according to the manufacturer's recommendations. The antibodies used in this study are listed in Supplementary Table 2.

**Tethering assays in human and S2 cells and RNA interference.** Tethering assays in human HEK293T cells using the β-globin reporter containing six MS2-binding sites (6xMS2bs)[33] were performed as previously described[8]. Briefly, cells were seeded in six-well plates ($0.8 \times 10^6$ cells per well) and transfected using Lipofectamine 2000 (Thermo Fisher Scientific).

The transfection mixtures contained 0.5 μg of the control plasmid (containing the β-globin gene fused to a fragment of the GAPDH gene but lacking MS2-binding sites)[33], 0.5 μg of the β-globin-6xMS2bs reporter and varying amounts of pcDNA3.1-MS2-HA plasmids expressing MS2-HA-tagged proteins. The plasmid amounts were as follows: for *Hs* Roquin1, 0.6 μg full length and 0.5 μg each Roq1-N and Roq1-C; for *Hs* Roquin2, 1.5 μg full length, 0.6 μg Roq2-N and 0.3 μg Roq2-C; for *Dm* Roquin, 0.3 μg full length, 0.1 μg each Roq-N and Roq-C, 0.2 μg each Roq-ΔCBM and ΔNBM and 0.3 μg Roq-ΔCBM + NBM. In the experiment shown in Fig. 2a,b, 0.175 μg of *Dm* Roquin, 0.4 μg of *Hs* Roquin1 and 1 μg of *Hs* Roquin2 were transfected. The transfection mixtures also contained plasmids expressing GFP (0.2 μg) or a GFP-tagged catalytically inactive DCP2 mutant (1.5 μg), as indicated.

When the β-globin-TNF-α reporter was used, cells were co-transfected with 0.5 μg β-globin-TNF-α reporter (CDE37 wild-type or MUT16; ref. 10) and plasmids expressing the HA-MS2 tagged proteins (*Hs* Roquin1: 0.5 μg full-length, 0.4 μg each Roq1-N and Roq1-C; *Dm* Roquin: 0.2 μg full-length, 0.1 μg each Roq-N, Roq-C, ΔCBM and ΔNBM and 0.2 μg ΔCBM + ΔNBM). In the

experiments shown in Fig. 8c–e, the cells were transfected with 0.05 µg of *Dm* Roquin and *Dm* Roquin M4. Cells were harvested 2 days after transfection.

For the λN-tethering assay in *Dm* S2 cells, $2.5 \times 10^6$ cells per well were seeded in six-well plates and transfected using Effectene (Qiagen). The transfection mixtures contained 0.1 µg firefly luciferase reporter (F-Luc-5BoxB, F-Luc-5BoxB-V5 or F-Luc-5BoxB-A$_{95}$C$_7$-HhR), 0.4 µg *Renilla* luciferase transfection control and various amounts of plasmids expressing λN-HA-tagged *Dm* Roquin constructs (0.01 µg full-length, 0.003 µg Roq-N, 0.025 µg Roq-C, 0.05 µg GST-Roq-CN, and 0.02 µg GST-Roq-CC). In the experiment described in Fig. 7f,g and Supplementary Fig. 9c–e, half of these amounts were transfected. In the experiment described in Supplementary Fig. 3g,h, the GST tags were replaced by GFP tags and 0.02 µg GFP-Roq-CN, and 0.005 µg GFP-Roq-CC were transfected. Cells were harvested 3 days after transfection.

NOT1 knockdowns using dsRNA were performed as previously described[35]. To measure the mRNA half-life, cells were treated with actinomycin D (5 µg ml$^{-1}$ final concentration) 3 days after transfection and collected at the indicated time points. RNA samples were analysed by northern blotting. The level of reporter mRNA was normalized to the levels of endogenous *rp49* mRNA in three independent experiments, averaged and plotted against time. The data were fitted to a double exponential decay function prior to averaging. The reported half-lives ($t_{1/2}$) correspond to 50% decay with respect to the initial amount of reporter RNA. Half-life errors are calculated from the standard fitting error.

Knockdowns in HeLa cells (provided by O. Mühlemann) were performed as described previously[2]. The 19 nt target sequences are as follows: control 5′-ATT CTCCGAACGTGTCACG-3′, CAF40 5′-GATCTATCAGTGGATCAAT-3′. Cells were transfected in six-well plates using Lipofectamine 2000 according to the manufacturer's protocol. Transfection mixtures contained 0.2 µg of *Dm* Roq-C; 0.3 µg of *Hs* Roq1-C and 0.4 µg of *Hs* Roq2-C.

Firefly and *Renilla* luciferase activities were measured using a Dual-Luciferase Reporter Assay system (Promega). Northern blotting was performed as previously described[35].

**Protein expression and purification.** All recombinant proteins were expressed in *E. coli* BL21 (DE3) Star cells (Invitrogen) in LB medium at 20 °C overnight. *Dm* Roquin fragments were expressed as fusion proteins containing N-terminal MBP tags cleavable by HRV3C protease[48]. In addition, the *Dm* Roq-C, -CN and -CC fragments carried an HRV3C-cleavable C-terminal GB1-6xHis tag[49]. The cells were lysed using an Avestin Emulsiflex-C3 homogenizer in lysis buffer (50 mM HEPES-NaOH (pH 7.5), 200 mM NaCl, 20 mM imidazole and 2 mM β-mercaptoethanol) supplemented with complete EDTA-free protease inhibitors (Roche), 5 µg ml$^{-1}$ DNaseI and 1 mg ml$^{-1}$ lysozyme. The proteins were separated from the crude lysate using amylose resin (New England Biolabs) and subsequently eluted from the resin in lysis buffer containing 25 mM D-(+)-maltose. The proteins containing a GB1-6xHis tag were further purified by nickel affinity chromatography using a HiTrap IMAC column (GE Healthcare). Proteins without GB1-6xHis tags were further purified by anion exchange chromatography using a HiTrapQ column (GE Healthcare). The final purification step for all proteins was size exclusion chromatography using a Superdex 200 16/600 column (GE Healthcare) in a buffer containing 10 mM HEPES-NaOH (pH 7.5), 200 mM NaCl and 2 mM DTT.

*Hs* CAF40 (residues 19–285) was expressed with a 6xHis tag cleavable by HRV3C protease. Lysis was carried out in lysis buffer containing 50 mM potassium phosphate (pH 7.5), 500 mM NaCl, 10% glycerol, 20 mM imidazole and 2 mM β-mercaptoethanol supplemented with complete EDTA-free protease inhibitors, DNaseI and lysozyme. The *Hs* 6xHis-CAF40 ARM domain was isolated from the lysate using a HiTrap IMAC column (GE Healthcare). The 6xHis tag was removed by cleavage using HRV3C protease during dialysis in low salt buffer containing 50 mM Tris-HCl (pH 7.5), 150 mM NaCl and 1 mM DTT. Subsequently, *Hs* CAF40 was further purified using a HiTrap Heparin column (GE Healthcare) followed by gel filtration on a Superdex 200 26/600 column (GE Healthcare) in gel filtration buffer containing 20 mM Tris-HCl (pH 7.5), 150 mM NaCl and 1 mM DTT.

*Dm* CAF40 (residues 25–291) was expressed with an N-terminal MBP tag. The cells were lysed in a buffer containing 50 mM HEPES-NaOH (pH 7.5), 500 mM NaCl and 2 mM DTT supplemented with complete EDTA-free protease inhibitors, DNaseI and lysozyme. The protein was isolated from the crude lysate using amylose resin followed by anion exchange chromatography using a HiTrapQ column (GE Healthcare). The MBP tag was removed by cleavage with HRV3C protease during dialysis in a buffer containing 10 mM potassium phosphate (pH 8.0), 400 mM NaCl, 10% glycerol, 50 mM ammonium sulfate and 2 mM DTT. The final purification step was size exclusion chromatography using a Superdex 200 26/600 column equilibrated with the same buffer.

The pentameric human NOT1-2-3-7-9 complex was obtained by co-expression of NOT1 (residues 1093–2371), MBP-NOT2-C, 6xHis-NOT3-C, GST-NOT7 and 6xHis-CAF40-ARM. The cells were lysed in buffer containing 50 mM potassium phosphate (pH 7.5), 300 mM NaCl and 2 mM DTT supplemented with complete EDTA-free protease inhibitors, DNaseI and lysozyme. The complex was purified using amylose resin, and the tags were removed by cleavage with HRV3C protease during dialysis in buffer containing 20 mM HEPES-NaOH (pH 7.5), 100 mM NaCl, 5% glycerol and 2 mM DTT. The complex was further purified using a HiTrap Heparin column (GE Healthcare) followed by size exclusion chromatography on a Superdex 200 26/600 column equilibrated in 20 mM HEPES-NaOH (pH 7.5), 300 mM NaCl, 5% glycerol and 2 mM DTT.

To purify the complex containing 6xHis-NOT1-MIF4G (residues 1093–1317) bound to MBP-CAF1, cells coexpressing the proteins were lysed in a buffer containing 50 mM HEPES-NaOH (pH 7.5), 300 mM NaCl, 20 mM imidazole and 2 mM β-mercaptoethanol supplemented with complete EDTA-free protease inhibitors, DNaseI and lysozyme. The complex was isolated from the crude lysate using amylose resin followed by chromatography on a HiTrap IMAC column. The affinity tags were removed by cleavage using the HRV3C protease, followed by final size exclusion chromatography on a Superdex 200 26/600 column in a buffer containing 10 mM HEPES-NaOH (pH 7.5), 200 mM NaCl and 2 mM DTT.

The NOT1 CD (residues 1607–1815) was expressed with an N-terminal MBP tag. The cells were lysed in a buffer containing 50 mM HEPES (pH 7.5), 300 mM NaCl and 2 mM DTT supplemented with complete EDTA-free protease inhibitors, DNaseI and lysozyme. The protein was isolated from the lysate using amylose resin followed by a HiTrapQ column. The MBP tag was removed by cleavage with HRV3C protease, followed by final size exclusion chromatography on a Superdex 200 26/600 column using a buffer containing 10 mM HEPES-NaOH (pH 7.5), 200 mM NaCl and 2 mM DTT.

Purification of the *Hs* NOT module comprising the NOT1-SHD (residues 1833–2361), NOT2-C (residues 350–540) and NOT3-C (residues 607–748) and the complex comprising NOT1-CN9BD (residues 1356–1588) bound to the CAF40-ARM has been previously described[2,9].

The *Dm* Roquin CBM peptide (residues 790–810) used for crystallization was synthesized by EMC Microcollections. The peptide was dissolved in 20 mM Tris-HCl (pH 7.5), 150 mM NaCl and 1 mM DTT.

**Crystallization.** *Dm* Roquin CBM was mixed with the purified *Hs* CAF40 ARM domain in an equimolar ratio. Initial crystallization screens were carried out using the sitting drop vapor diffusion method at 22 °C by mixing 200 nl of the CAF40-Roquin complex solution (at 6.75 mg ml$^{-1}$) with 200 nl of reservoir solution. Crystals appeared within 1 day under several conditions. Optimized crystals grew at 18 °C after 1 day using hanging drops after mixing 0.8 µl of the CAF40-Roquin complex solution (at 2.25 mg ml$^{-1}$) with 0.8 µl of reservoir solution containing 0.1 M NaOAC (pH 5.0), 17.5% (w v$^{-1}$) PEG 4000 and 0.1 M AmSO$_4$. Crystals were cryoprotected using reservoir solution supplemented with 15% glycerol and flash-frozen in liquid nitrogen.

**Data collection and structure determination.** Diffraction data were collected on a PILATUS 6M detector (Dectris) at the PXII beamline of the Swiss Light Source. The best data set extending to a resolution of 2.15 Å was recorded at a wavelength of 1.000 Å and processed in space group P2$_1$ using XDS and XSCALE[52]. Two copies of the CAF40 ARM domain (PDB-ID 2FV2, chain A) were found in the asymmetric unit by molecular replacement using PHASER[53] from the CCP4 package[54]. This initial model was improved by iterative cycles of model building in COOT[55] and refinement using PHENIX[56]. As the final step, two Roquin CBM peptides were modeled into the density and improved by further refinement cycles.

The final model was refined with excellent stereochemistry to R$_{work}$ = 18.4% and R$_{free}$ = 22.6% and includes all residues of the two CAF40 molecules (residues 19–285 plus six residues from the HRV3C cleavage site and linker sequences; chains A and C) and all residues (790–810) of the Roquin peptides for both chains (B and D). For the C-terminal two pairs of α-helices in CAF40, the final atomic B-factors are clearly above average, pointing to an elevated mobility and/or statistical disorder in this part of the molecule, which is not involved in binding the CBM.

***In vitro*** **MBP-pulldown assays.** Purified MBP (20 µg) or MBP-tagged Roquin fragments (40 µg) were incubated with equimolar amounts of purified CCR4-NOT subcomplexes and 50 µl of the amylose resin slurry (New England Biolabs) in 1 ml of pulldown buffer (50 mM HEPES-NaOH (pH 7.5), 200 mM NaCl, 2 mM DTT). After a 1-h incubation, the beads were washed five times with pulldown buffer and the proteins were eluted with pulldown buffer supplemented with 25 mM D-(+)-maltose. The eluted proteins were precipitated with trichloroacetic acid and analysed by SDS-PAGE followed by Coomassie blue staining.

**Data availability.** The coordinates for the structure of the *Dm* Roquin CBM peptide bound to CAF40 were deposited in the Protein Data Bank (PDB) under ID code 5LSW. The authors declare that the data supporting the findings of this study and relevant source data are available within the article and its Supplementary Information file. Other data and materials are available from the authors upon request.

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

## Acknowledgements

We thank Dr Oliver Mühlemann for the kind gift of the HeLa cell line used in knockdown experiments and Dr Jens Lykke-Andersen for providing the plasmids for performing MS2 tethering assays in human cells. We are grateful to Heike Budde, Sigrun Helms, Maria Fauser and Catrin Weiler for excellent technical support. We thank the staff at the PX beamlines of the Swiss Light Source for assistance with data collection. This work was supported by the Max Planck Society.

## Author contributions

A.S. and P.B. cloned proteins for expression in human and S2 cells. A.S. conducted tethering assays in human and Drosophila cells and co-immunoprecipitation assays in Drosophila cells with help from D.K.-O. and D.B. P.B. conducted pulldown assays in

human cells. T.R. and A.S. expressed and purified proteins from *E. coli* for crystallization and pulldown assays, crystallized the CAF40-CBM complex, conducted *in vitro* pulldown assays and analysed the data. T.R. solved and refined the crystal structure of the CAF40-CBM complex. Y.C. cloned and purified the pentameric complex. O.W. supervised the structural part of the study. E.I. was the principal investigator and conceived and supervised the project. A.S., T.R., O.W. and E.I. wrote the manuscript with contributions from P.B. and D.K.-O.

## Additional information

**Competing financial interests:** The authors declare no competing financial interests.

