## [Peer Review File · Nature Communications]

Reviewers' Comments:

Reviewer #1 (Remarks to the Author):

Sgromo et al. present a structure/functional analysis of the protein Roquin, which binds specific mRNAs and recruits the CCR4-NOT deadenylase complex to cause mRNA degradation and regulate post-transcriptional gene expression. Vertebrate Roquins regulate genes involved in helper T cell differentiation and are important in autoimmunity.

It was already known that Hs Roquin1 recruits CCR4 through its C-terminal region. The main finding here is that both Dm Roquin and Hs Roquin2 also recruit CCR4-NOT (via at least the CAF40 subunit) through direct interactions from their C-terminal regions, despite not being conserved with the corresponding C-terminal region of Hs Roquin1. A crystal structure of a minimal Dm Roquin peptide (CBM) necessary and sufficient for the interaction, bound to CAF40 is presented.

Overall, the paper is very well written, clearly laid out and very thorough in its identification and characterization of interactions. However, the principles of CCR4 NOT recruitment are similar to that observed with other proteins and in this regard this is a somewhat incremental finding. The CBM-CAF40 interaction is well characterized and robust, however, the NBM results are less convincing, particularly with regards to discrepancies between activity and binding of the RoqCN fragment. The authors might consider focusing the manuscript even more around the CBM-CAF40 interaction. Specific comments/criticisms are outlined below.

1) Comparing Fig. 1e to Fig. 2a. In Fig. 1e, the amount mRNA degradation by Hs Roquin2 is about 85%, but in Fig. 2a, the amount of mRNA degradation by Hs Roquin2 is only 70% (with GFP control). I would have expected these numbers to match more closely given the tight error bars in Fig. 1e. The same is true for Dm Roquin (Fig. 1h vs. Fig. 2a). This might be explained by protein levels of Hs Roquin2, but Dm Roquin levels are about the same in both experiments.

2) Fig. 3d: The results of RoqCC binding appear robust and agree with activity measurements in Supp. Fig. 3b. However, what does not appear to make sense is that RoqCN loses substantial binding for NOT1,2,3 compared to RoqC (Fig. 3d lane 21). Yet, it retains almost as much activity as RoqC in Supp. Fig. 3b. The activity of RoqCN is also better than RoqCC, which actually has the robust interaction with CAF40.

3) Suppl. Fig. 3 For Roq CC and CN activity measurements, GST is being used as a tag here. Could the dimerization of GST artificially accentuate the activities measured for these fragments?

4) pg. 11: “However, only the Roq-CC fragment, comprising residues 702–819, bound to the purified CAF40 protein in vitro, whereas the Roq-CN fragment (residues 501–702) did not.

Fig. 3d lane 20 does show weak binding of RoqCN to CAF40 (greater than MBP control lanes) and comparable to what is characterized later as weak binding for RoqCN and NOT fragments.

5) Supp. Fig. 3g – A role for direct interaction with NBM is not yet convincing, since removing it from RoqCC could be affecting the structure of the remaining fragment, particularly since NBM on its own does not appear to have even weak residual binding to NOT fragments.

6) Fig. 4b and 4c vs. Supp. Fig. 3g. The results don't appear to agree here: In Supp. Fig. 3g lane 12 RoqCC robustly pulls down and binds NOT1,2,3 and deltaNMB knocks this binding out. But in 4b and 4c there is essentially no binding of Dm Roq full-length to NOT2 or NOT3 (given that 1% of input is loaded while 30% of IP is loaded) and this is completely unaffected by deleting NBM, once again questioning whether NBM has any direct binding role.

7) pg. 14 “Finally, the CBM peptide is fixed by a salt bridge between N804 and the CAF40 residue Q98 (helix $\alpha 5$) and by a hydrogen bond from CAF40 N88 (helix $\alpha 5$) to the carbonyl oxygen of G791 (Fig. 5g).”

The interaction between N804 and Q98 is hydrogen bonding, not a salt-bridge

8) Either Fig. 6b and 6c need to swapped, or their Figure calls on pg. 15.

9) Average B-factors are high for a 2.15 Angstrom structure. What does the Wilson plot indicate? Additionally, the protein B-factors are higher than water B-factors which is unusual. Does ligand refer to the CBM peptide? If so, B-factors of over 100 for it at this resolution appears to be an error? I would include a difference electron density map for the peptide in the supplementary section.

Reviewer #2 (Remarks to the Author):

In this study, Sgromo et al. characterize in detail the interaction of the drosophila melanogaster (dm) Roquin protein with CAF40, a subunit of the CCR4-NOT deadenylase complex though

which Roquin mediates the degradation of target mRNAs. The authors first delineate a short motif (CBM) at the C-terminal end of dm Roquin that interacts with CAF40. They then solve the structure of the CBM peptide in complex with human (hs) CAF40 by crystallography, which allows them to identify the contacting residues in both Roquin-CBM and CAF40.

This is a well written manuscript that combines in vitro interaction studies with in cellulo co-IP experiments, functional assays for Roquin activity and structural elucidation of the dm Roquin-CAF40 interaction. The experiments are carried out very carefully, and I have only minor concerns regarding the presented data. However, since the authors show that Roquin contacts the CCR4-NOT complex via different motifs (at least two in dm Roquin), my main concern pertains to the functional relevance of the Roquin-CAF40 interaction. Addressing this aspect would strengthen the manuscript.

Major comment:

How important is CAF40 for association of dm and hs Roquin proteins with the CCR4-NOT complex, and how important is CAF40 for the activity of dm and hs Roquin? CAF40 knock down experiments would allow the authors to clarify this point, and help to make the case that the Roquin-CAF40 interaction is indeed functionally important.

Minor comments:

Fig. 1f,i,l: Since the "control" mRNA lacking the MS2bs is longer than the 6xMS2bs transcript, it must contain additional sequence. Please include a description of the control transcript.

Fig. 1k and l: The two panels appear alphabetically in the wrong order.

Fig. 2i: Interestingly, the authors observe that tethering of dm Roquin suppresses protein expression of a reporter mRNA that lacks a terminal poly-A tail (A95-C7-Hhr), while tethering does not induce decay of this reporter mRNA. The authors suggest that this result is compatible with recruitment of the CCR4-NOT complex. It would be more convincing if the authors could provide evidence for this hypothesis by conducting the same tethering assays under NOT1 knockdown conditions.

Page 15, first and second paragraph: Fig 6b and 6c appear to be mixed up.

Fig. 6d is confusing: Which protein was immunoprecipitated? The anti-GFP label suggests that GFP-CAF40 (wt and mutants) were IPed, but the blot suggests that Roquin/F-luc were IPed. Is CAF40 HA- or GFP-tagged?

MPI f. Developmental Biology · Spemannstr. 35/II · D-72076 Tübingen

Spemannstr. 35/II
D-72076 Tübingen
Germany

Dr. E. Izaurralde
Tel.: +49 (0)7071-601-1350
Fax: +49 (0)7071-601-1353
elisa.izaurralde@tuebingen.mpg.de

July 23th, 2014

NCOMMS-16-17638

Response to reviewer 1

The reviewer states that: “Overall, the paper is very well written, clearly laid out and very thorough in its identification and characterization of interactions. (...) The CBM-CAF40 interaction is well characterized and robust, however, the NBM results are less convincing” and he/she suggested that we “might consider focusing the manuscript even more around the CBM-CAF40 interaction”.

We agree with the reviewer and have modified the manuscript accordingly.

Response to specific comments:

1) Comparing Fig. 1e to Fig. 2a. In Fig. 1e, the amount mRNA degradation by Hs Roquin2 is about 85%, but in Fig. 2a, the amount of mRNA degradation by Hs Roquin2 is only 70% (with GFP control. I would have expected these numbers to match more closely given the tight error bars in Fig. 1e. The same is true for Dm Roquin (Fig. 1h vs. Fig. 2a). This might be explained by protein levels of Hs Roquin2, but Dm Roquin levels are about the same in both experiments.

The experiments in Fig. 1e and 2a cannot be compared because different amounts of plasmids have been transfected. We have added additional information in the Materials and Methods to clarify the differences.

2) Fig. 3d: The results of RoqCC binding appear robust and agree with activity measurements in Supp. Fig. 3b. However, what does not appear to make sense is that RoqCN loses substantial binding for NOT1,2,3 compared to RoqC (Fig. 3d lane 21). Yet, it retains almost as much activity as RoqC in Supp. Fig. 3b. The activity of RoqCN is also better than RoqCC, which actually has the robust interaction with CAF40.

The *in vitro* pulldown experiments with purified components and *in vivo* tethering experiments are not expected to show a 1:1 correlation. Tethering is very sensitive even when the protein retains only low binding affinity. Additionally, it is possible that *in vivo*, the Roq-CN recruits other factors that stabilize the interaction and / or mediate repression or decay. We mention this possibility in the revised manuscript.

3) Suppl. Fig. 3 For Roq CC and CN activity measurements, GST is being used as a tag here. Could the dimerization of GST artificially accentuate the activities measured for these fragments?

To exclude that the GST tag and a possible dimerization perturb the results, we repeated the experiments with a GFP tag. The results with the GFP tag are comparable to the results with the GST tag and the corresponding panels are included in Supplementary Fig. 3g,h.

4) pg. 11: “However, only the Roq-CC fragment, comprising residues 702–819, bound to the purified CAF40 protein *in vitro*, whereas the Roq-CN fragment (residues 501–702) did not.

We agree with the reviewer and have corrected the statement on page 11 to account for the fact that Roq-CN binding is strongly reduced but not fully abolished.

5) Supp. Fig. 3g – A role for direct interaction with NBM is not yet convincing, since removing it from RoqCC could be affecting the structure of the remaining fragment, particularly since NBM on its own does not appear to have even weak residual binding to NOT fragments.

The Roq-CC fragment is predicted to be unstructured. This argues against an indirect effect of the NBM via the CBM and rather supports a direct interaction. But probably the CBM and

additional residues are required to stabilize the interaction. As suggested by the reviewer, we modified the text to tone down the role of the NBM.

6) Fig. 4b and 4c vs. Supp. Fig. 3g. The results don't appear to agree here: In Supp. Fig. 3g lane 12 RoqCC robustly pulls down and binds NOT1,2,3 and deltaNMB knocks this binding out. But in 4b and 4c there is essentially no binding of *Dm* Roq full-length to NOT2 or NOT3 (given that 1% of input is loaded while 30% of IP is loaded) and this is completely unaffected by deleting NBM, once again questioning whether NBM has any direct binding role.

Roq-CC binding in the IP is present and specific (see the control lane). The fact that the NBM and CBM deletions have little effect in cells probably relates to additional uncharacterized contacts of the full length *Dm* Roq or stabilization by additional binding partners.

7) pg. 14 “Finally, the CBM peptide is fixed by a salt bridge between N804 and the CAF40 residue Q98 (helix α 5) and by a hydrogen bond from CAF40 N88 (helix α 5) to the carbonyl oxygen of G791 (Fig. 5g).” The interaction between N804 and Q98 is hydrogen bonding, not a salt-bridge.

We thank the reviewer for pointing out this mistake and have corrected the text.

8) Either Fig. 6b and 6c need to swapped, or their Figure calls on pg. 15.

This has been corrected.

9) Average B-factors are high for a 2.15 Angstrom structure. What does the Wilson plot indicate? Additionally, the protein B-factors are higher than water B-factors which is unusual. Does ligand refer to the CBM peptide? If so, B-factors of over 100 for it at this resolution appears to be an error? I would include a difference electron density map for the peptide in the supplementary section.

The Wilson B-factor is 38.7 \AA^2 . The somewhat elevated average atomic B-factor of 60.0 \AA^2 is primarily a consequence of the two C-terminal pairs of α -helices in the two CAF40 molecules, which are much less well defined in the electron density than the remainder of the structure.

Thus, the N-terminal two thirds of the CAF40 molecules have an average B-factor of 45.1 Å², whereas the C-terminal third has an average B-factor of 88.4 Å² and no discernable water molecules surrounding it. This C-terminal third is not involved in binding the CBM. The Roquin CBM peptides have an average B-factor of 59.0 Å² and are well defined in the electron density. As suggested by the reviewer, we now included two additional panels in the Supplementary Fig. 7. Supplementary Fig. S7e shows the difference density for the CBM before the CBM was modelled. Supplementary Fig. S7f shows the density for the CBM in an unbiased simulated annealing composite omit map.

Additionally, in Table 1, we replaced 'Ligands' by 'Other solvent molecules'. These are four sulphate ions and two putative Tris molecules, which we modelled with full occupancy. The high B-factors should be taken as an indication, however, that the respective sites are probably not fully occupied.

Response to reviewer 2

The referee states that “this is a well-written manuscript that combines *in vitro* interaction studies with *in cellulo* co-IP experiments, functional assays for Roquin activity and structural elucidation of the *Dm* Roquin-CAF40 interaction. The experiments are carried out very carefully, and I have only minor concerns regarding the presented data”.

Major comment:

How important is CAF40 for association of *Dm* and *hs* Roquin proteins with the CCR4-NOT complex, and how important is CAF40 for the activity of *Dm* and *hs* Roquin? CAF40 knock down experiments would allow the authors to clarify this point, and help to make the case that the Roquin-CAF40 interaction is indeed functionally important.

We have performed CAF40 knockdown experiments as suggested by the reviewer. Depletion of CAF40 in S2 cells does not suppress Roquin activity. This result is consistent with the observation that deletion of the CBM from *Dm* Roquin had no detectable effect in tethering assays. Therefore this experiment was not included in the paper. In contrast, CAF40 depletion in HeLa cells suppressed the activity of *Hs* Roquin1, Roquin2 and *Dm* Roquin by 2–2.5 fold. This experiment is included in Supplementary Fig. 5a–c. These results indicate that CAF40 is indeed an important recruitment factor, but other, redundant interactions compensate for the lack of CAF40 in *Dm* and human cells.

Minor comments:

Fig.1f,i,l: Since the "control" mRNA lacking the MS2bs is longer than the 6xMS2bs transcript, it must contain additional sequence. Please include a description of the control transcript.

As stated in the Material and Methods, the control mRNA reporter contains the β -globin ORF fused to a portion of the GAPDH gene but lacks MS2 binding sites and is therefore longer than the 6xMS2bs transcript. We added a sentence in the Methods. These reporters have been described in (ref. 32).

Fig.1k and l: The two panels appear alphabetically in the wrong order.

We thank the reviewer for detecting this mistake, which has been corrected.

Fig.2i: Interestingly, the authors observe that tethering of dm Roquin suppresses protein expression of a reporter mRNA that lacks a terminal poly-A tail (A95-C7-Hhr), while tethering does not induce decay of this reporter mRNA. The authors suggest that this result is compatible with recruitment of the CCR4-NOT complex. It would be more convincing if the authors could provide evidence for this hypothesis by conducting the same tethering assays under NOT1 knockdown conditions.

We agree with the reviewer that the results are consistent with this hypothesis but we cannot rule out alternative mechanisms because even after NOT1 depletion a residual activity is observed. Therefore, we have rephrased this sentence.

Page 15, first and second paragraph: Fig 6b and 6c appear to be mixed up.

We thank the reviewer for noticing this mistake, which has been corrected.

Fig.6d is confusing: Which protein was immunoprecipitated? The anti-GFP label suggests that GFP-CAF40 (wt and mutants) were IPed, but the blot suggests that Roquin/F-luc were IPed. Is CAF40 HA- or GFP-tagged?

We agree with the reviewer that the labels in the figure were not clear and have modified the figure accordingly.

Reviewers' Comments:

Reviewer #1 (Remarks to the Author):

The authors have appropriately addressed all of my comments and the revised manuscript has been improved.

Reviewer #2 (Remarks to the Author):

No further comments